# Elucidation of the anti-autophagy mechanism of the *Legionella* effector RavZ using semisynthetic LC3 proteins

**Aimin Yang[1,2,3†], Supansa Pantoom[2,3†], Yao-Wen Wu[1,2,3*]**

[1]Institute of Chemical Biology and Precision Therapy, Zhongshan School of Medicine, Sun Yat-Sen University, Guangzhou, China; [2]Chemical Genomics Centre of the Max Planck Society, Dortmund, Germany; [3]Max-Planck-Institute of Molecular Physiology, Dortmund, Germany

**Abstract** Autophagy is a conserved cellular process involved in the elimination of proteins and organelles. It is also used to combat infection with pathogenic microbes. The intracellular pathogen *Legionella pneumophila* manipulates autophagy by delivering the effector protein RavZ to deconjugate Atg8/LC3 proteins coupled to phosphatidylethanolamine (PE) on autophagosomal membranes. To understand how RavZ recognizes and deconjugates LC3-PE, we prepared semisynthetic LC3 proteins and elucidated the structures of the RavZ:LC3 interaction. Semisynthetic LC3 proteins allowed the analysis of structure-function relationships. RavZ extracts LC3-PE from the membrane before deconjugation. RavZ initially recognizes the LC3 molecule on membranes via its N-terminal LC3-interacting region (LIR) motif. The RavZ $\alpha$3 helix is involved in extraction of the PE moiety and docking of the acyl chains into the lipid-binding site of RavZ that is related in structure to that of the phospholipid transfer protein Sec14. Thus, *Legionella* has evolved a novel mechanism to specifically evade host autophagy.

*For correspondence: yaowen. wu@mpi-dortmund.mpg.de

†These authors contributed equally to this work

Competing interests: The authors declare that no competing interests exist.

## Introduction

Autophagy is an evolutionarily conserved intracellular degradation process in eukaryotes, which is essential for cellular homeostasis in response to various environmental and cellular stresses. During autophagy, a double-membrane structure, termed as phagophore (or isolation membrane) engulfs the cytosolic components and closes to form an autophagosome, which subsequently fuses with a lysosome, resulting in degradation of internal contents (*Mizushima and Komatsu, 2011*). Autophagosome formation is the key process in autophagy. The biogenesis of autophagosomes is believed to be dependent on the formation of lipidated Atg8. Microtubule-associated protein light chain 3 (LC3) and GABARAP family proteins are the orthologs of yeast Atg8 in mammalian cells. Lipidated and membrane-associated Atg8/LC3 has been used as a bona fide marker of autophagosomes and progression of autophagy (*Fujita et al., 2008*; *Sou et al., 2008*). Production of lipidated Atg8/LC3 is controlled by two ubiquitin-like conjugation systems. Newly synthesized Atg8/LC3 is processed by a protease, Atg4, to expose a C-terminal glycine. The resulting Atg8/LC3 is conjugated to phosphatidylethanolamine (PE) through a ubiquitin-like conjugation reaction controlled by Atg7, Atg3 and the Atg12-Atg5:Atg16 complex. The Atg12-5:16 complex is generated by another ubiquitin-like conjugation system controlled by Atg7 and Atg10. Atg4 releases lipidated Atg8/LC3 from the surface of autophagosomes (*Kimura et al., 2007*; *Kirisako et al., 2000*; *Tanida et al., 2004*; *Xie et al., 2008*). However, the role of Atg8-PE and its regulators in autophagosome formation remain poorly understood.

**eLife digest** Organisms have to fight bacteria and other disease-causing microbes on a daily basis. To stay on top of the game, cells use many ways to defend themselves. Autophagy, for example, is a process that breaks down unwanted or damaged molecules inside cells, which has also been linked to fighting infections caused by disease-causing microbes. During this process, structures called autophagosomes engulf the molecules or microbes, and digest them with the help of enzymes.

Forming an autophagosome is a complex process that requires several steps and molecules. First, cells create a growing membrane sac that collects the debris and eventually seals to form the autophagosome. One of the key proteins that expands the membrane is the protein LC3, which is located on the both sides of the autophagosome's membrane. LC3 must be linked to an oily molecule (or lipid), known as PE for short, in order to interact with this membrane. The combined molecule LC3-PE then provides a docking station for receptor proteins that collect and deliver the debris, including microbes, into the growing autophagosome. A specific part of these receptors called LIR, short for LC3-interacting region motif, connects the receptors to LC3.

Some bacteria have evolved mechanisms to avoid autophagy or can even hijack the process to survive in the host cell. For example, the bacterium *Legionella pneumophila*, which causes Legionnaire's disease, manipulates the main molecular pathways involved in autophagy to avoid being digested by the host cell. The bacterium injects a protein called RavZ into the cell, which splits the lipid component from LC3-PE. It was still unknown, however, how RavZ can recognize and split LC3-PE.

Yang et al. created LC3 proteins with different PE fragments to study the molecular pathways underlying this process. The experiments revealed that RavZ also contains LIR motifs that it uses to recognize and attach to LC3. After RavZ binds, it extracts LC3-PE from the membrane of autophagosomes and integrates the PE part into its own lipid-binding site. RavZ then splits LC3-PE by removing the lipid component of the protein. Building on this knowledge, Yang et al. were able to experimentally prevent RavZ from breaking up LC3-PE. This suggests that hindering RavZ from binding by blocking its LIR motif could represent a potential pharmaceutical approach to stop *L. pneumophila* from avoiding autophagy.

A next step will be to confirm if blocking RavZ could indeed support autophagy. It will also be useful to find out if other microbes use the same mechanisms as *L. pneumophila* to avoid autophagy.

Autophagy also serves as a defense mechanism against invading pathogens (termed xenophagy) (*Deretic, 2011*; *Levine, 2005*). Xenophagy recognizes bacteria through autophagy receptors that contain two crucial domains, the ubiquitin-binding domain (UBD) and LC3-interacting region (LIR) motifs, which are important for cargo recognition and interaction with the LC3 proteins, respectively. The receptor binds to the ubiquitinated pathogen through its UBD and recruits it to the autophagosome membrane via the interaction of the LIR motifs to LC3 proteins (*Kirkin et al., 2009*; *Korac et al., 2013*; *Mostowy et al., 2011*; *Thurston et al., 2009*; *von Muhlinen et al., 2012*; *Wild et al., 2011*). LIR motifs are also present in receptor and scaffold proteins involved in other selective autophagy processes and play an essential role in recruiting components of the autophagy machinery to phagophores (*Klionsky and Schulman, 2014*; *Noda et al., 2010*). However, it is not clear how LIR motifs selectively recognize mammalian Atg8 family members.

Some bacteria have evolved specific mechanisms to avoid autophagy or even hijack the autophagy machinery in order to survive in the cell (*Choy and Roy, 2013*). The pathogenic bacterium *Legionella pneumophila*, which causes Legionnaire's disease, manipulates the core machinery of autophagosome formation to evade host autophagy. *L. pneumophila* inhibits autophagy by injecting an effector protein called RavZ into the cytoplasm. RavZ functions as a cysteine protease and irreversibly deconjugates mammalian Atg8s from PE to inhibit autophagosome formation (*Choy et al., 2012*). Unlike Atg4 that cleaves the amide bond between terminal glycine and PE, RavZ cleaves the amide bond before glycine. As a consequence, the RavZ-cleaved Atg8 proteins cannot be

relipidated, leading to inhibition of autophagy. RavZ represents an interesting pathogenic effector, functional characterization of which will shed light on the mechanism of autophagosome biogenesis.

To date, how RavZ recognizes and deconjugates LC3-PE is not known. This is largely due to the previously insurmountable difficulties in recombinant preparation and handling of lipidated LC3 proteins. Herein, semi-synthetic LC3 proteins make it possible to elaborate the mechanism of RavZ function. We have used chemical methods to produce LC3 proteins with different C-terminal modifications, enabling the analysis of structure-function relationships of LC3-deconjugation by RavZ, allowing formulation of a membrane extraction model. We find that RavZ extracts LC3-PE from membranes and then deconjugates C-terminal Gly-PE. We show that the second N-terminal LIR motif (LIR2) is required for RavZ activity and RavZ:LC3 interaction. The crystal structures of RavZ: LC3 and LIR2:LC3 complexes and interaction analysis suggest RavZ initially recognizes LC3 mainly via its LIR2 motif. We identify the lipid-binding site (LBS) of RavZ, which shows a similar fold to that of the LBS of yeast phospholipid transfer proteins (Sec14 family). The LBS involves a highly dynamic and hydrophobic helix α3 that is engaged in association with the membrane and plays an essential role in extraction of the conjugated PE from the membrane. Therefore, by a combination of chemical, biophysical and cell biological approaches, we gain insights into a novel mode of host-pathogen interaction.

## Results

### Semi-synthesis of LC3 proteins with various C-terminal modifications

Advances in protein ligation methods have provided a powerful tool for studying post-translational modified proteins (*Dawson and Kent, 2000*; *Hackenberger and Schwarzer, 2008*; *Vila-Perelló and Muir, 2010*). In order to address the mechanism of *Legionella* effector RavZ function in host autophagy, we sought to produce LC3 proteins with various C-terminal modifications by expressed protein ligation (EPL) and direct aminolysis of protein thioesters (*Figure 1*). Previously, we reported the semi-synthesis of lipidated protein LC3-PE using a combination of lipidated peptide synthesis and EPL (*Yang et al., 2013*). An MBP-assisted solubilization strategy was used to facilitate ligation under folding conditions and to solubilize the lipidated protein without detergents and membranes.

The C-terminal peptides containing DPPE (16:0) (1,2-dipalmitoyl-*sn*-glycero-3-phosphoethanolamine), DHPE (6:0) (1,2-dihexanoyl-*sn*-glycero-3-phosphoethanolamine) and 1-hexadecanol (C16) were ligated with MBP-LC3$^{1–114}$-thioester (*Figure 1—figure supplement 1* and *Supplementary file 1*). Direct aminolysis strategy (*Payne et al., 2008*; *Yi et al., 2010*) was used to produce LC3 proteins with modification of the soluble PE fragments, including ethanolamine (EA, **1**), phosphoethanolamine (pEA, **2**), glycerophosphoethanolamine (GpEA, **8**) and diacetyl glycerophosphoenthanolamine (DAGpEA, **10**) (*Figure 1*; *Supplementary file 2* and *Supplementary file 3*).

### Structure-function relationship study of LC3-deconjugation by RavZ

First of all, we examined whether RavZ can act on semisynthetic LC3-PE. RavZ only cleaved LC3-PE but not pro-LC3, whereas Atg4 cleaved both substrates (*Figure 2—figure supplement 1A*). In keeping with previous studies, Atg4B treatment led to LC3$^{1–120}$, whereas RavZ-mediated deconjugation resulted in LC3$^{1–119}$ (*Figure 2—figure supplement 1B*). In a previous report, RavZ deconjugated LC3-PE from membranes (*Choy et al., 2012*). However, our results suggest that RavZ can cleave LC3-PE without the requirement for membranes.

The question then arises as to how RavZ recognizes LC3-PE. Based on the results shown above, two possible hypotheses could be made. First, RavZ would recognize a certain soluble fragment derived from the PE rather than the lipid chain, which could not be 'seen' by RavZ when it is buried in the membrane (*Figure 2A*). Second, the fatty acid chain would be required for binding to RavZ. In this case, LC3-PE would have to be extracted from the membrane by RavZ before proteolytic cleavage could occur. To distinguish these scenarios, a structure-function relationship study of RavZ-mediated deconjugation is required (*Figure 2B*). The semisynthetic LC3 proteins make it possible to perform such an analysis. LC3 proteins with different C-terminal modifications were subjected to RavZ and Atg4B treatment. The measurements showed that RavZ cannot cleave LC3 proteins containing soluble fragments derived from PE, while Atg4B is active toward these proteins (*Figure 2—figure supplement 2A*). RavZ only cleaves PE-modified LC3 proteins, with a preference for long fatty

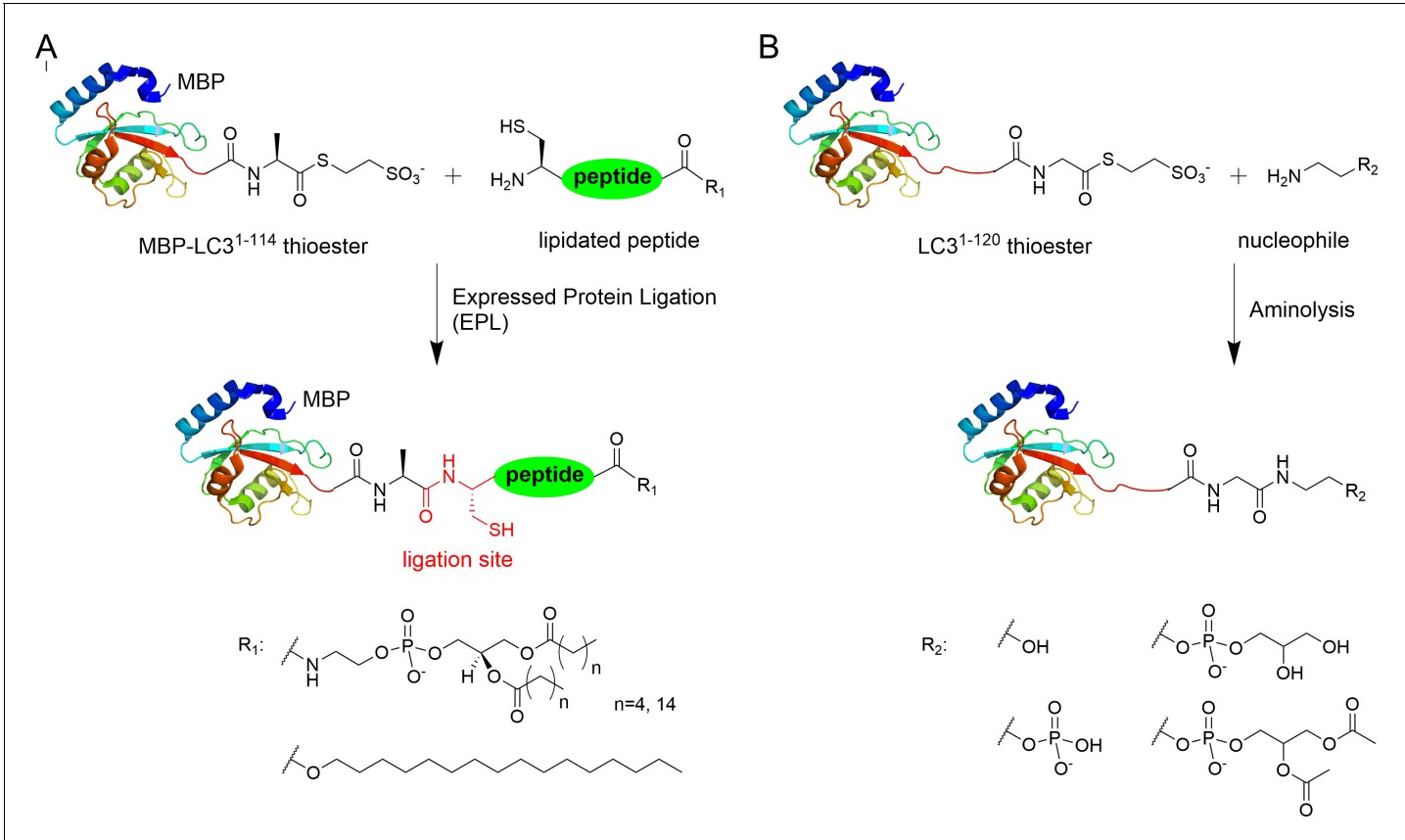

**Figure 1.** Semisynthesis of LC3 proteins with various C-terminal modifications. (**A**) Strategy for the semisynthesis of lipidated LC3 proteins using expressed protein ligation (EPL). (**B**) Strategy for the semisynthesis of LC3 proteins containing soluble fragments of PE by direct aminolysis.
The following figure supplement is available for figure 1:

**Figure supplement 1.** Synthesis of peptides and PE fragments.

acid chain. Enzyme kinetic data show that the catalytic efficiency of RavZ for LC3-PE (16:0) ($k_{cat}/K_m$ = 1140 M$^{-1}$·s$^{-1}$) is 12 times higher than that for LC3-PE (6:0) ($k_{cat}/K_m$ = 93.8 M$^{-1}$·s$^{-1}$) (*Figure 2B*; *Figure 2—figure supplement 2B–D*). LC3-C16 cannot be processed by RavZ, suggesting that LC3 with a single fatty acid chain is not sufficient for hydrolysis by RavZ (*Figure 2—figure supplement 2E*). In contrast, Atg4B can cleave all modified LC3 proteins tested (*Figure 2C*; *Figure 2—figure supplement 2A and E*). Taken together, in contrast to Atg4B activity that does not require any specific structure downstream of the C-terminal glycine, RavZ activity is strictly dependent on conjugated PE structures. Therefore, it is conceivable that RavZ contains a lipid-binding site and can extract LC3-PE from the membrane before cleavage. This extraction model is further confirmed in later experiments.

Interestingly, RavZ did not process PE-peptides containing 6, 7 or 11 C-terminal amino acid residues. To further confirm the findings, we ligated the CQETFG-PE peptide to the C-terminus of Rab7 GTPase. Again, RavZ cannot hydrolyze the chimeric Rab7-PE protein (*Figure 2—figure supplement 1C*). These results suggest that a C-terminal PE-modified peptide is not sufficient for substrate recognition and cleavage by RavZ.

## RavZ extracts LC3-PE from membranes

The structure-function relationship studies prompted us to examine the extraction model of RavZ. To this end, we used protease-deficient RavZ$^{C258A}$ in an LC3-PE extraction assay (*Figure 3A*). MCF-7 cells stably expressing GFP-LC3 were subjected to starvation to induce autophagy. The membrane

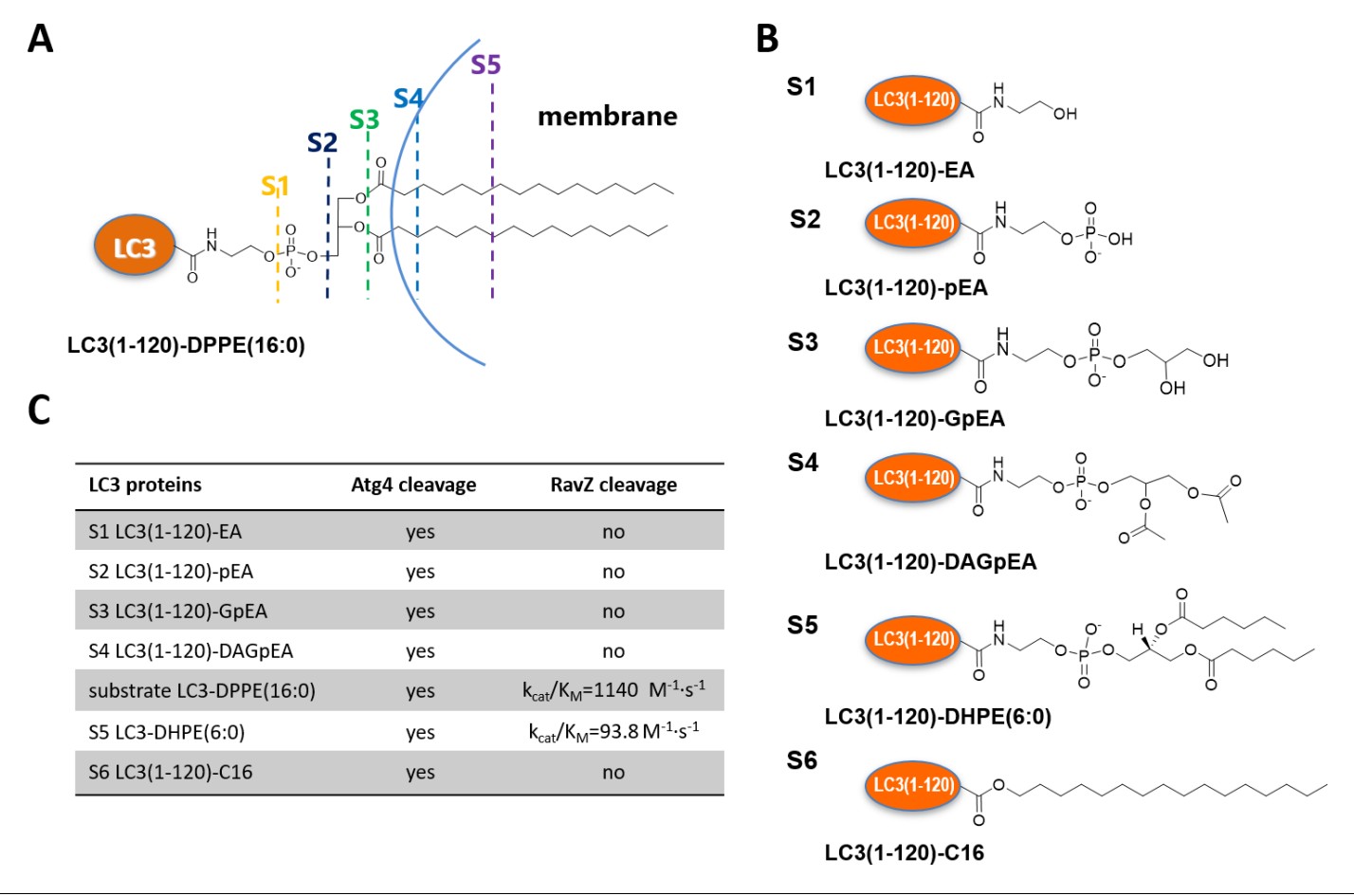

**Figure 2.** Structure-function relationship of LC3-deconjugation by RavZ. (**A**) Schematic diagram of proteo-membrane containing LC3-PE. The fatty acid chains of PE are buried in the lipid bilayer, serving as the membrane anchor. (**B**) LC3 proteins containing different fragments derived from the PE. Ethanolamine, EA; phosphoethanolamine, pEA; glycerophosphoethanolamine, GpEA and diacetyl glycerophosphoenthanolamine, DAGpEA. (**C**) In vitro cleavage of semisynthetic LC3 proteins by Atg4B and RavZ.

The following figure supplements are available for figure 2:

**Figure supplement 1.** In vitro cleavage of LC3 proteins with various C-terminal modifications.

**Figure supplement 2.** Structure-function relationship study of LC3-deconjugation by RavZ.

fraction of cells was incubated with different concentrations of RavZ[C258A] protein. The supernatant was precipitated using TCA/DOC (trichloroacetic acid/sodium deoxycholate). The soluble proteins in supernatant and the membrane-associated proteins were visualized by immunoblotting with anti-LC3 antibody. Our results showed that RavZ[C258A] treatment decreased the level of membrane association of both endogenous LC3-II (lipidated LC3) and GFP-LC3-II in a dose-dependent manner. Accordingly, the level of soluble LC3-II and GFP-LC3-II increased with increasing concentration of RavZ[C258A]. These results suggest that RavZ[C258A] extracts lipidated LC3 from membranes.

To further evaluate the extraction activity of RavZ in vivo, GFP-LC3 stable cells were transfected with mCherry, mCherry-RavZ wt or mCherry-RavZ[C258A]. Both RavZ wt and RavZ[C258A] significantly inhibited GFP-LC3 puncta formation (*Figure 3B and C*). However, the inhibitory effect of RavZ[C258A] was not observed in a previous report, where GFP-LC3 and RavZ[C258A] were transiently expressed in the cell (*Choy et al., 2012*). We repeated the experiment using cells transiently expressing GFP-LC3. We also find that RavZ wt but not RavZ[C258A] inhibits GFP-LC3 puncta formation under these

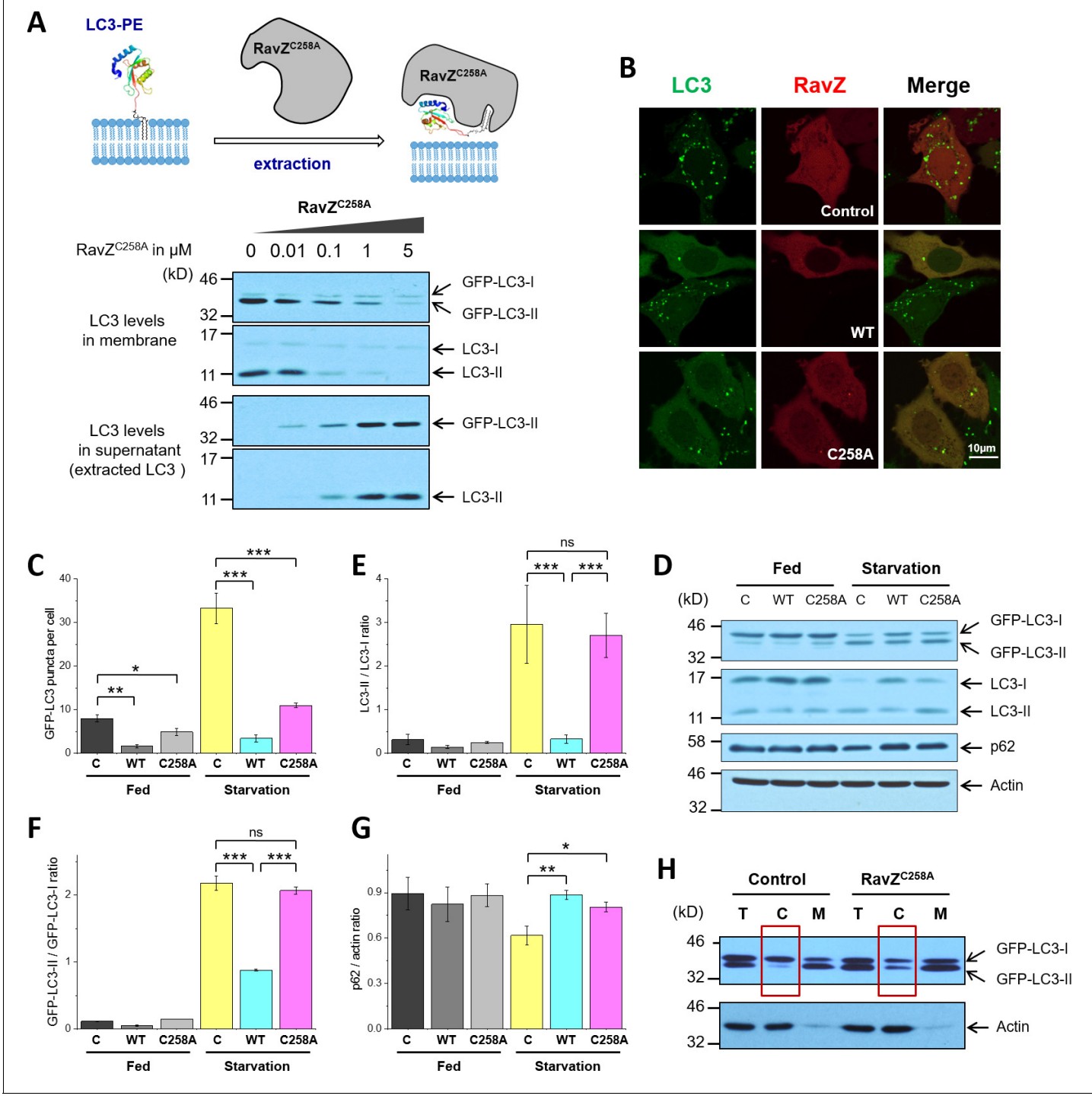

**Figure 3.** RavZ extracts LC3-PE from membranes. (**A**) LC3-PE extraction assay. Endogenous LC3-PE and GFP-LC3-PE on membranes from GFP-LC3 stable cells (20 µg) were treated with increasing concentrations of recombinant RavZ^C258A. After incubation for 2 hr at 37°C, the membrane and soluble fractions were separated by ultracentrifuge and subjected to western blotting with LC3-specific antibody. Upper panel shows the schematic diagram of LC3-PE extraction by RavZ^C258A. (**B**) Confocal microscopy of GFP-LC3 stable cells transfected with mCherry-RavZ constructs. Cells were treated for 2 hr under starvation medium (EBSS). A complete set of images for GFP-LC3 cells transfected with all mCherry-RabZ constructs (WT, C258A, and their mutants) are shown in *Figure 6—figure supplement 2A and B*. (**C**) Quantification of GFP-LC3 puncta shown in (**B**). n = 21–32 cells. Mean and SD are presented; ***p<0.001, *p<0.05. (**D**) Effect of RavZ on LC3-II level in cells. GFP-LC3 stable cells were transfected with wild-type RavZ (WT) and RavZ^C258A and cultured either in growth or in starvation medium. (**E–G**) Quantification of ratio of endogenous LC3-II to LC3-I (**E**), GFP-LC3-II to GFP-LC3-I (**F**) and p62 to actin (**G**) in (**D**). n = 3 independent experiments. Mean and SD are presented; ***p<0.001, *p<0.05, ns: not significant. (**H**) Validation of LC3-PE

*Figure 3 continued on next page*

*Figure 3 continued*

extraction by RavZ in the cell. GFP-LC3 stable cells were transfected with RavZ[C258A]. After starvation, cells were lysed. Membrane fractionation was performed by ultracentrifuge. The membrane and soluble fractions were subjected to western blotting.

The following figure supplement is available for figure 3:

**Figure supplement 1.** Membrane binding and α3 are required for extraction activity of RavZ.

conditions, probably due to the high expression level of GFP-LC3 in transient cells (*Figure 3—figure supplement 1A–1E*). RavZ wt but not RavZ[C258A] led to decrease in LC3-II level in GFP-LC3 cells (*Figure 3D–3F*). Moreover, lipidated LC3 was found in the soluble fraction in RavZ[C258A]-expressing cells, which is in contrast to the control, where lipidated LC3 is only found in the membrane fraction (*Figure 3H*). Both RavZ wt and RavZ[C258A] significantly inhibited p62 degradation, a marker for autophagy flux (*Figure 3D and G*). Therefore, the inhibitory effect of RavZ[C258A] is a result of extraction of lipidated LC3 molecules from membranes, leading to accumulation of LC3-II in the cytosol and reduction of membrane-localized LC3 proteins. This scenario is quite different from the effect of overexpression of Atg4B[C74A] (a protease-deficient mutant), which inhibits LC3 lipidation by sequestration of unlipidated LC3 through formation of stable complexes in the cytosol (*Fujita et al., 2008*). Since the lipidated LC3 level is not changed in RavZ[C258A]-expressing cells, RavZ[C258A] inhibits autophagy by sequestering lipidated LC3 proteins rather than unlipidated LC3 proteins. These findings indicate that RavZ does not interfere with autophagy machinery upstream of Atg8/LC3 but specifically processes lipidated LC3 proteins.

## The LIR motif is essential for RavZ activity

The importance of the LIR motif for recognition of Atg8/LC3 in selective autophagy prompted us to identify LIR motifs in RavZ. The LIR motif is composed of conserved W/Y/FxxL/I/V sequence. The binding of LIR motifs with Atg8/LC3 proteins is quite conserved, with two key hydrophobic residues playing an essential role in interaction with the hydrophobic pocket of the Atg8s (*Ichimura et al., 2008*; *Noda et al., 2008*, *2010*). Three potential LIR motifs of RavZ were identified by the iLIR online server (*Kalvari et al., 2014*) (*Figure 4A*). The first and second LIR motifs are located at the N-terminal region of RavZ (residues 14–19, LIR1; and 27–32, LIR2), while the third motif is in the C-terminal region (residues 433–438, LIR3). To identify which LIR motif is involved in RavZ activity and to map the catalytic domain of RavZ, RavZ was dissected into fragments and subjected to the in vitro LC3-PE cleavage assay and to measurement of binding with LC3 by a fluorescence polarization assay (*Figure 4A*; *Figure 4—figure supplement 1B*). Truncation of LIR1 (20–502) and LIR3 (1–431) did not influence RavZ activity and led to only up to ca. twofold reduction of binding affinity with LC3, whereas simultaneous truncation of LIR1 and LIR2 (33–502 and 55–487) completely abolished cleavage activity and reduced the binding affinity with LC3 by 2–4 fold (*Figure 4A*; *Figure 4—figure supplement 1B*). Mutation of the key aromatic residue (F29A) of LIR2 led to dramatic reduction of RavZ activity, whereas mutation of LIR1 (F16A) did not affect RavZ activity. The RavZ[F16/F29A] mutant showed the same effect as RavZ[F29A] (*Figure 4C and D*). These results demonstrate that LIR2 is crucial for RavZ activity. However, fragments containing one of the LIR motifs can still form stable complexes with LC3. In contrast, RavZ fragments (55–430 and 55–331) without LIR motifs completely lost their binding ability to LC3 (*Figure 4A*; *Figure 4—figure supplement 1A and B*). Therefore, LIR2 is crucial for both binding and cleavage activities. Indeed, RavZ fragment 20–331 that contains only LIR2 can cleave LC3-PE with only twofold decrease in binding affinity to LC3. Thus, RavZ[20-331] represents the minimal catalytic domain. Prompted by these results, we tested binding of the LIR2 peptide (DIDEFDLLEGDE) to LC3 by ITC and binding of the fluorescein-labeled LIR2 peptide to LC3 by fluorescence polarization, leading to dissociation constants ($K_d$) of 360 nM and 550 nM, respectively, which is comparable with that of RavZ:LC3 complex ($K_d$ = 260 nM) (*Figure 4—figure supplement 1C and D*). Therefore, the LIR2 motif of RavZ contributes the major binding energy for RavZ:LC3 interaction.

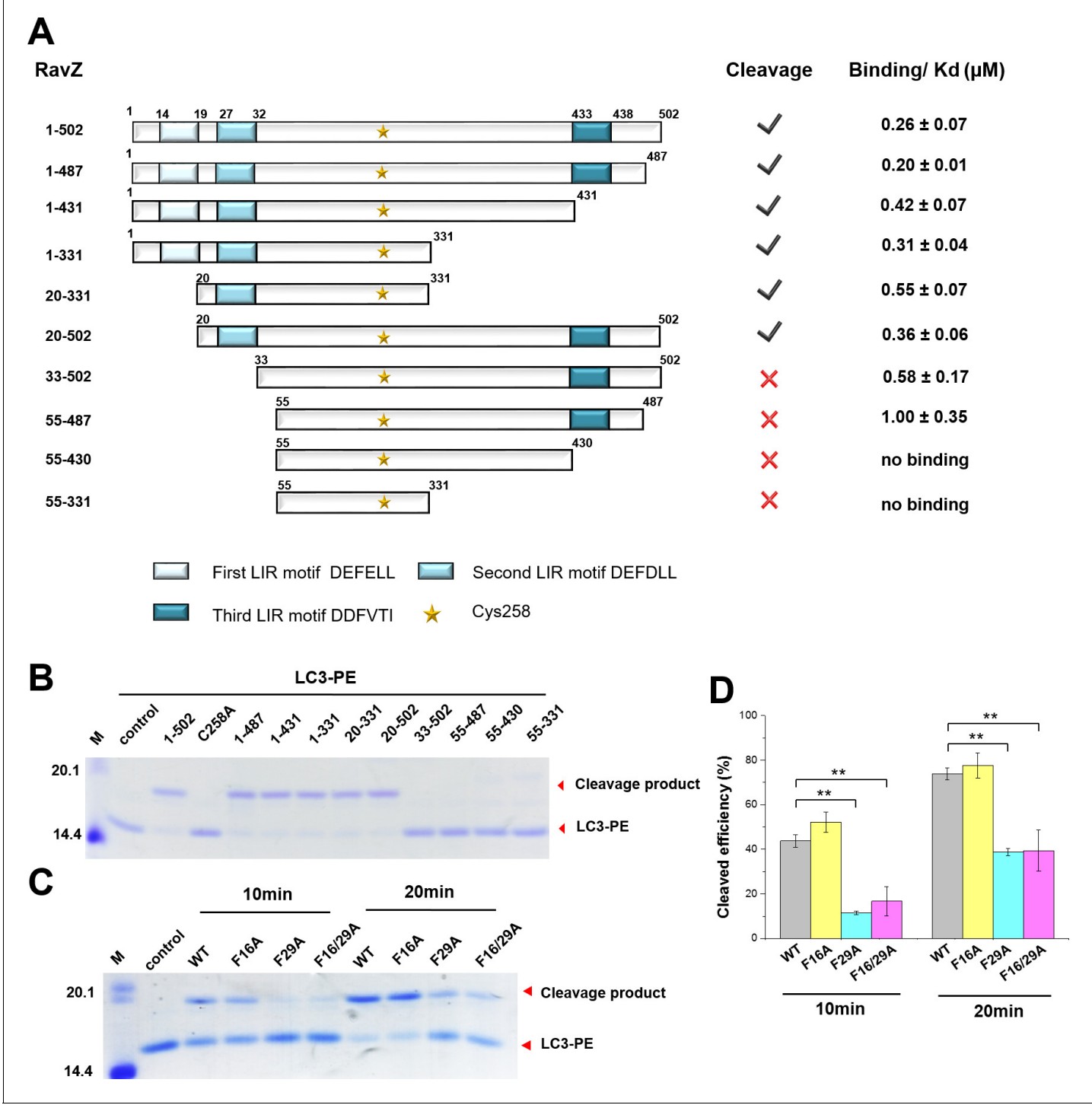

**Figure 4.** The LIR motif is essential for RavZ:LC3 binding and cleavage activity of RavZ. (**A**) Schematic diagram of RavZ fragments showing LIR motifs and the catalytic residue Cys258. Binding and cleavage activities of the fragments are shown. (**B**) In vitro LC3-PE cleavage assay of the RavZ fragments. (**C**) In vitro LC3-PE cleavage assay of RavZ containing LIR mutations. The LC3-PE protein (7 µM) was treated with RavZ mutants (700 nM) at 37°C for 10 min or 20 min and then resolved by SDS-PAGE. (**D**) Quantification of the LC3-PE cleavage in (**C**). n = 3 independent experiments. Mean and SD are presented; **p<0.01.

The following figure supplement is available for figure 4:

**Figure supplement 1.** LIR2 is required for RavZ:LC3 interaction.

## RavZ membrane binding is required for extraction

Next, we solved the structures of two RavZ fragments, 1–487 and 20–502. Crystal structures of the fragments are essentially identical, solved in space group I422. RavZ structures of both fragments are resolved from residues 47 to 433 and 48 to 432, respectively, whereas the missing regions at the N- and C-termini (residues 1–47 and 433–502) might be flexible and disordered. Our RavZ structures are identical to the reported RavZ structure (PDB:5CQC) with an RMSD of 0.33 (*Figure 5—figure supplement 2A*) (*Horenkamp et al., 2015*). RavZ shows two major domains, the N-terminal catalytic domain (residues 49–320, catalytic triad: C258, H176, D197) and the C-terminal domain (329-423). In line with the crystal structure, RavZ$^{1–331}$ showed similar LC3-binding affinity and catalytic efficiency to the wild-type RavZ (*Figure 4A*; *Figure 2—figure supplement 2F*, *Figure 2—figure supplement 2G* and *Figure 4—figure supplement 1B*), suggesting the catalytic domain is sufficient for deconjugation of LC3-PE. The C-terminal domain has been recently shown to be involved in association with phosphatidylinositol 3-phsphate (PI3P) and be essential for membrane binding (*Horenkamp et al., 2015*). Interestingly, although the catalytic domain is sufficient for cleavage, the extraction activity is reduced (*Figure 3—figure supplement 1F and G*). Since the catalytic domain does not bind to membranes (*Horenkamp et al., 2015*), membrane binding of RavZ is required for extraction of LC3-PE in vivo.

## Structural analysis of RavZ:LC3 interaction

To understand how RavZ recognizes LC3 before extraction, we solved the crystal structure of the RavZ$^{1–431}$:LC3$^{1–119}$ complex at 2.5 Å (*Supplementary file 4*). The complex shows 1:1 stoichiometry. However, the complex displays only a relatively small binding interface (300 Å$^2$). Nevertheless, the structure of RavZ in the complex shows a well-defined resolution of flexible loops, which are missing in the free-structure. The loop (249–253), loop (278–287) and the α8-α9 loop (346–356) are now visible (*Figure 5—figure supplement 2B*). Importantly, the N-terminal loop containing the LIR2 motif (residues 25 to 48) is resolved in the structure. LIR2 of RavZ interacts with the α2-β3 loop of LC3 instead of the canonic LIR-binding site that involves hydrophobic pockets located in β1, β2 and α3 (*Noda et al., 2008*, *2010*). Further analysis of the crystal contacts between symmetry-related RavZ molecules (RavZa and RavZb) shows that LIR2 of RavZa interacts extensively with RavZb, involving the α3 loop and the β6-β7 loop (*Figure 5A*). Comparing with the free-RavZ structure, the α3 loop in the complex moves ca. 9 Å inward, suggesting a dynamic nature of this region (*Figure 5B*). Key hydrophobic interactions include F29 (RavZa) with Y211 (RavZb), L31 (RavZa) with I170 (RavZb), L208 (RavZb) and F212 (RavZb). It is conceivable that the crystal packing led to a low-energy binding configuration involving interaction of LIR2 with the α3 loop of its symmetric mate. Such an interaction is less likely to exist outside the crystal, because the affinities of the α3 mutant (Y211D/F212D/Y216D) and the wild-type RavZ toward LC3 are identical (*Figure 6—figure supplement 2C*). The result indicates that there is no defined LC3-binding site on RavZ except for the LIR2 loop.

To prove whether LIR2 interacts with the LIR-binding site in LC3, we solved two structures of an LIR2-LC3 fusion in two crystallization conditions. Both structures are identical and solved in the same space group (*Supplementary file 4*). There are two molecules (molecule A and B) of the LIR2-LC3 fusion in one asymmetric unit with LIR2 of each molecule interacting with its symmetry-related LC3 molecule (*Figure 5—figure supplement 2D*). The hydrophobic interaction of LIR2 with LC3 is conserved and adopts the classical LIR-LC3-binding mode (*Klionsky and Schulman, 2014*; *Noda et al., 2010*). The first aromatic residue, F29, of LIR2 interacts with the first hydrophobic pocket (HP1) consisting of I23, P32, L53, F108 and the alkyl sidechain of K51 and the second aromatic residue L32 interacts with the second hydrophobic pocket (HP2) consisting of F52, V54, P55, V58, L63 and I66 (*Figure 5C and D*). The salt bridge interactions of residues LIR2$^{E28}$ with LC3$^{K51}$ and LIR2$^{D30}$ with LC3$^{R70}$ are observed in both structures. We further investigate whether RavZ shows preference on binding to individual Atg8 family proteins, we tested binding of LIR2 peptide with three Atg8 members (representing two subfamilies), LC3B, GABARAPL1 and GATE16. No significant difference in binding affinities was observed (*Figure 5—figure supplement 1A*). Therefore, RavZ has no preference in binding to Atg8 proteins.

To verify the interaction of LIR2 with LC3 seen in the crystal structure, we mutated the interacting residues on LC3, hydrophobic residues (F52 and L53) and charged residues (K51 and R70), to alanine. The mutants were subjected to binding analysis with the LIR2 peptide by fluorescence

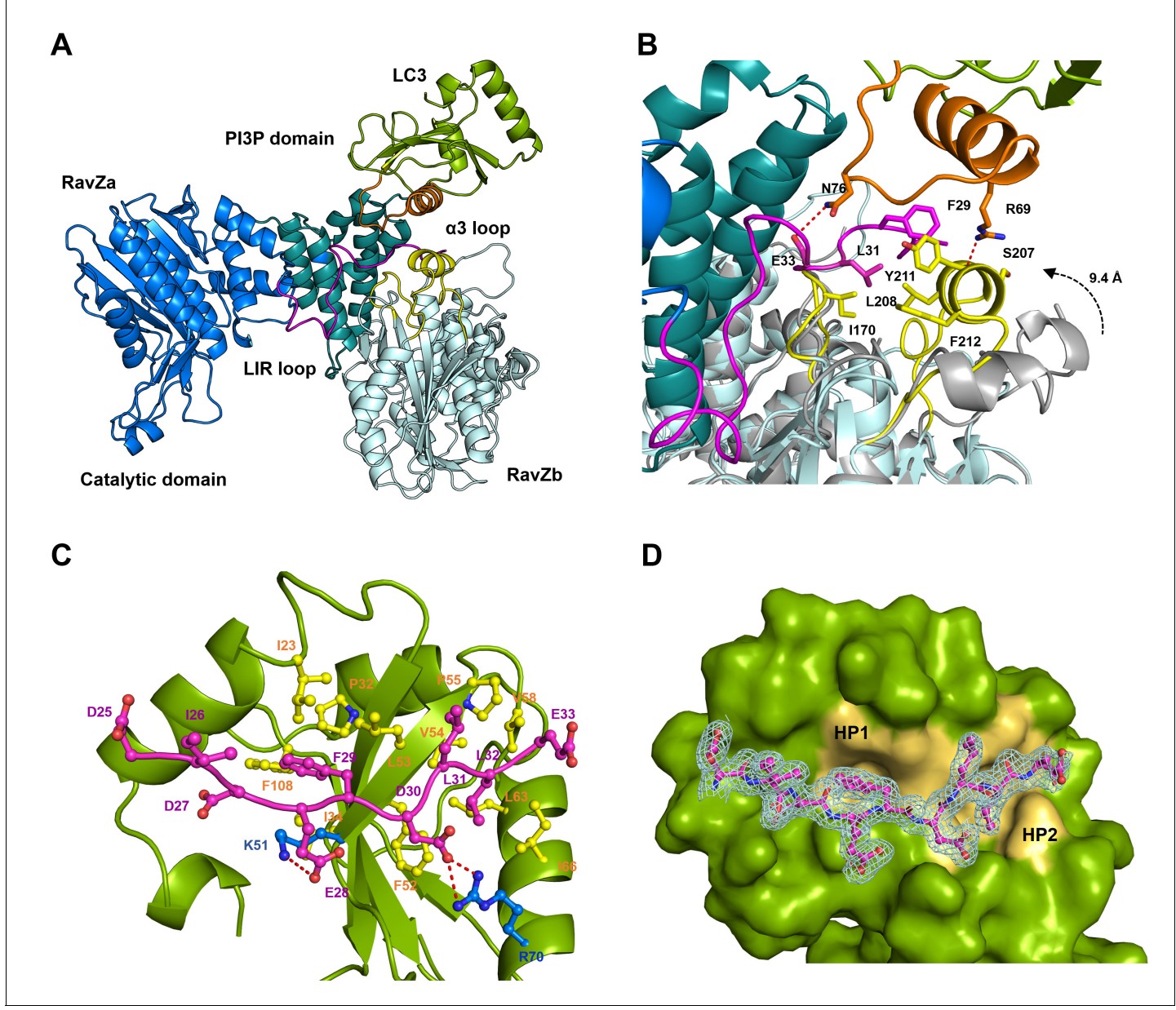

**Figure 5.** Structural basis of RavZ:LC3 interaction. (**A**) Crystal structure of the RavZ:LC3 complex. Catalytic domain, the PIP3-binding domain and the N-terminal loop containing LIR2 of RavZa are shown in blue, teal and pink, respectively. The LC3 molecule is shown in green with its α2 and the α2–β3 loop colored in orange. The symmetry-related RavZ (RavZb) molecule is shown in pale cyan with its α3 and the the β6–β7 loop colored in yellow. (**B**) Binding interface between LC3 with RavZa and RavZb. Free RavZ structure (gray) is superimposed with RavZb to show the conformational change of α3. Hydrogen bonds are shown as red dashed lines. (**C**) Crystal structure of the LIR2:LC3 complex. LC3 is colored in green and the LIR2 peptide is coloured in pink. Hydrogen bonds are shown as red dashed lines. (**D**) Interaction of LIR2 with LC3. LIR peptide is shown as pink stick and LC3 is displayed as surface. Two hydrophobic binding pockets (HP1 and HP2) are shown in pale yellow. The 2Fo–Fc map of the LIR2 motif is contoured density at α of 1.0.

The following figure supplements are available for figure 5:

**Figure supplement 1.** LIR2 is required for RavZ:LC3 interaction.

**Figure supplement 2.** Structural analysis of free RavZ and RavZ:LC3 complex.

polarization. F52A mutation led to the reduction of LIR2 binding by about sixfold ($K_d$ = 3.14 μM), while R70A mutation reduced binding affinity about 15-fold ($K_d$ = 8.10 μM). L53A and K51A mutation led to only residual effect on binding to LIR2. No binding of the free LIR2 peptide with the LIR2-LC3 fusion protein was observed under these conditions, suggesting that the LIR2 peptide competes with the conjugated LIR2 peptide for binding to the LIR-binding site of LC3 (*Figure 4—figure supplement 1C*). Moreover, RavZ and the LIR2 peptide bind to LC3 in a competitive manner, as shown by the competitive titration and enzymatic inhibition assay (*Figure 5—figure supplement 1B and C*). Therefore, RavZ recognizes LC3 mainly via its N-terminal LIR2 motif by binding to the LIR-binding site in LC3.

## Molecular basis of LC3-PE extraction by RavZ

In order to understand the thermodynamic basis of RavZ extraction, we sought to determine the binding affinity of RavZ$^{C258A}$ with LC3-PE and LC3$^{1-119}$ by microscale thermophoresis (MST) technique. To make LC3-PE soluble in solution without detergent, MBP tag was left intact. The measurements showed that RavZ$^{C258A}$ binds to MBP-LC3-PE and MBP-LC3$^{1-119}$ with dissociation constants (Kd) of 23 ± 4 nM and 69 ± 5 nM, respectively, which suggests that RavZ binds to lipidated LC3 in three times higher affinity than unlipidated LC3 (*Figure 6—figure supplement 1D*). Therefore, the thermodynamic driving force for RavZ extraction is modest but still favorable.

Based on our model, there should be a lipid-binding site (LBS) in RavZ. The amino acid sequence of RavZ does not give any hint of the LBS. We superimposed the free RavZ structure with the phospholipid-binding domain of Sec14 proteins (PDB: 1AUA and 3B74), which are the major yeast phosphatidylinositol (PtdIns)/phosphatidylcholine (PtdCho) transfer proteins and are essential for lipid metabolism (*Schaaf et al., 2008*; *Welti et al., 2007*). Structural alignment indicated that part of the N-terminal catalytic domain of RavZ (α2–α4, β6–β9) shows a similar fold to that of the core-lipid-binding domain of Sec14, consisting of 5 β-strands and 4 α-helices (α7–α10) (*Schaaf et al., 2008*) (*Figure 6A*). The β sheets of Sec14 serve as the floor of hydrophobic binding pocket, along helices α7-α10 that gate the Sec14 pocket, forming extensive van der Waals contacts with the bound fatty acid chains. Helices α9–α10 act as lid helices, which move toward helix α8 upon lipid binding to form a closed conformation to capture lipid into the binding site (*Figure 6—figure supplement 1C*) (*Welti et al., 2007*). Similarly in RavZ, helices α2 (corresponding to α7 of Sec14) and α4 (corresponding to α8 of Sec14) along β sheets form a large hydrophobic interface (*Figure 6A and B*). However, the hydrophobic pocket is closed in RavZ, suggesting that a conformational change may be required to accommodate the fatty acid chains. A lid helix in RavZ is not clearly observed based on the structural alignment. However, due to the highly dynamic and hydrophobic nature of α3, it is possible that α3 may resemble the function of the lid helix and may undergo substantial movements to interact and stabilize the lipid moiety in the LBS upon binding of PE.

In order to prove the predicted LBS of RavZ, we mutated a set of hydrophobic residues on α2, α3, α4, the α4-β9 loop and β7 to the negatively change residue Asp (*Figure 6B and C*). The In vitro LC3-PE cleavage assay showed that most of the mutants affect the cleavage activity of RavZ. The double mutation (Y211D/F212D, α3-m2) or triple mutation (Y211D/F212D/Y216D, α3-m3) in α3 completely abolished the cleavage activity. A single mutation scan shows that F211D, Y212D and L208D dramatically reduce the cleavage activity, whereas the Y216D mutant largely retains activity. Triple mutation of the gate residues on α4 (L224D/I288D/L232D) and the α4-β9 loop (F242D/L239D/F237D) led to significant reduction in cleavage activity. To confirm the floor residues of the hydrophobic pocket of RavZ, we mutated hydrophobic residues on α2 (L139D/L143D) and β7 (L180D/I182D). Again, these mutations significantly reduced cleavage activity (*Figure 6C*).

To evaluate the deconjugation activity of RavZ mutants in cells, GFP-LC3 cells were transfected with different mCherry-RavZ mutant constructs. An additional mutant RavZ$^{F163D}$ (F163 on β6 located inside of the hydrophobic cavity) was also included in the assay. The level of LC3-II was evaluated by Western blotting (*Figure 6D*) and GFP-LC3 puncta assay (*Figure 6F*). Cells expressing RavZ mutants displayed a significant increase in the ratio of endogenous LC3-II/LC3-I as well as the ratio of GFP-LC3-II/GFP-LC3-I and in the number of GFP-LC3 puncta, compared to those expressing wild-type RavZ (*Figure 6E and F*; *Figure 6—figure supplement 2A*), suggesting that deficient activity of RavZ mutants results in decreased deconjugation of LC3-PE in cells. Moreover, the α3-m2 mutant completely abolished RavZ deconjugation activity in vivo. These results are in line with those obtained by the in vitro LC3-PE cleavage assay.

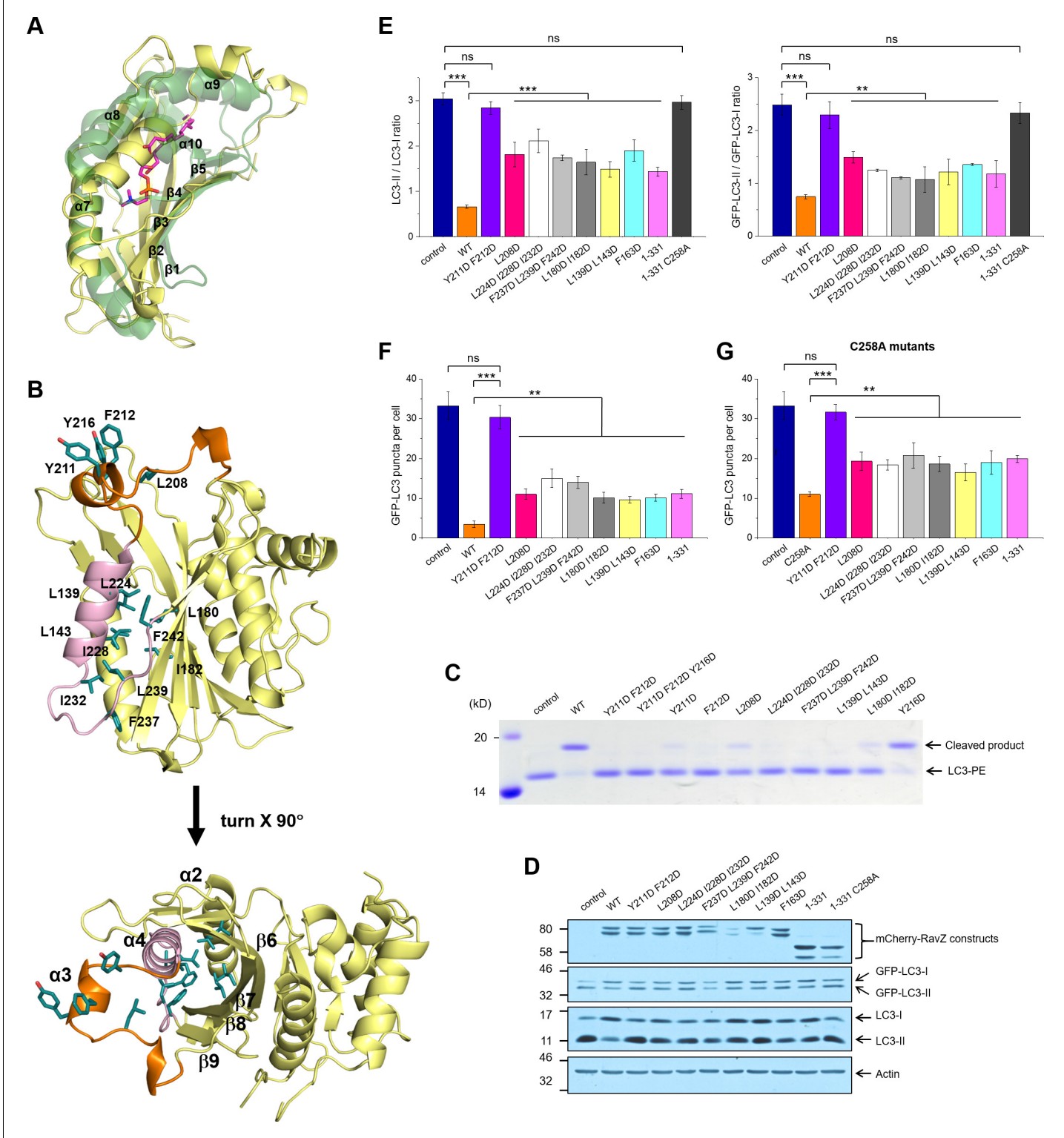

**Figure 6.** Identification of the lipid-binding site (LBS) of RavZ. (**A**) Structural alignment between the lipid-binding sites of yeast Sec14 homolog (Shf1) (PDB: 3B74) and RavZ. RavZ is colored in pale yellow, while Shf1 is shown as a green transparent cartoon and its secondary elements are indicated. The phosphatidylethanolamine (PE) is shown as stick model with carbon atoms colored in magenta. (**B**) Structure of N-terminal catalytic domain of RavZ showing the LBS of RavZ and the predicted lipid-binding residues. The α3 loop and α4 are colored in orange and light pink, respectively. The residues involved in lipid binding are highlighted and displayed as side chain sticks. (**C**) In vitro LC3-PE cleavage assay of RavZ mutants. (**D**) Effect of RavZ mutants on LC3-II level in the cell. GFP-LC3 stable cells were transfected with RavZ mutants and subjected for starvation for 2 hr. (**E**) Quantification of

*Figure 6 continued on next page*

*Figure 6 continued*

the ratio of endogenous LC3-II to LC3-I (left) and GFP-LC3-II to GFP-LC3-I (right) shown in (D). n = 3 independent experiments. Mean and SD are presented; ***p<0.001, **p<0.01, ns: not significant. (F) Effect of RavZ mutants on GFP-LC3 puncta formation in the cell as shown in *Figure 6—figure supplement 2A*. n = 25–37 cells. Mean and SD are presented; ***p<0.001, **p<0.01, ns: not significant. (G) Effect of RavZ$^{C258A}$ mutants on GFP-LC3 puncta formation in the cell as shown in *Figure 6—figure supplement 2B*. n = 22–35 cells. Mean and SD are presented; ***p<0.001, **p<0.01, ns: not significant.

The following figure supplements are available for figure 6:

**Figure supplement 1.** Structural analysis of the lipid-binding site of free RavZ, free Sec14 and PE-bound Sfh1.

**Figure supplement 2.** Functional characterization of the lipid-binding site of RavZ.

To further examine the extraction activity of RavZ mutants, GFP-LC3 cells were transfected with mCherry-RavZ$^{C258A}$ mutant constructs. In keeping with the deconjugation results, a significantly higher number of GFP-LC3 puncta was observed in cells expressing RavZ$^{C258A}$ mutants than those expressing RavZ$^{C258A}$, indicating less extraction activity of these mutants in vivo (*Figure 6G*; *Figure 6—figure supplement 2B*). Consistently, the α3-m2 and α3-m3 mutants completely lost extraction activity (*Figure 6G*; *Figure 3—figure supplement 1F and G*). Moreover, all RavZ mutants bind to LC3 with similar $K_d$ values to those of the wild-type protein (*Figure 6—figure supplement 2C*) and display similar secondary structure with comparable α helix content to the wild-type protein (*Figure 6—figure supplement 2D*). Therefore, the mutations do not significantly affect protein folding and binding to LC3. Taken together, we conclude that the LBS of RavZ, consisting of α2, α3, α4, α4–β9 loop and β sheets (β6–β9), accommodates the fatty acid chains of PE. The LBS plays an essential role in extraction and cleavage of LC3-PE.

## Discussion

In previous studies, the substrate LC3-PE was produced on liposomes by recombinant reconstitution of the ubiquitin-like conjugation system (i.e. mammalian Atg7, Atg3 and Atg8s) (*Choy et al., 2012*). The question then arises whether RavZ activity requires membranes or not. However, production of liposome-free LC3-PE and manipulation of LC3-PE structures are difficult using the recombinant approach. In this study, the semisynthetic approach makes it possible to address these inherent problems in the analysis of post-translationally modified proteins. We have used chemical approaches to prepare LC3 proteins with various modifications in a membrane-free manner.

Using semisynthetic LC3 proteins modified with various PE fragments, we analyzed the structure-function relationship of LC3 deconjugation by RavZ. We demonstrate that RavZ activity is strictly dependent on the lipid structure of the substrate and RavZ cleaves LC3-PE without requirement for membranes. These observations indicate an extraction model of RavZ function, which is further proved by the extraction assay using protease-deficient RavZ. This is the first-time identification of such an action mode for host-pathogen interactions. Although the N-terminal catalytic domain is sufficient to bind and cleave LC3-PE in solution, its extraction activity and thereby its deconjugating activity in vivo are attenuated (*Figure 6E–G*; *Figure 3—figure supplement 1F and G*). Therefore, association of RavZ with membranes via its C-terminal PI3P-binding domain is also required for efficient extraction in vivo, probably by targeting RavZ to the autophagosome membrane (*Horenkamp et al., 2015*). *L. pneumophila* may have evolved both functions simultaneously, that is extraction and cleavage. The former determines the specificity and the latter confers the turnover. Without retrieval of LC3-PE from membranes RavZ cannot perform proteolytic activity. However, although extraction alone is sufficient to inhibit autophagy, it requires a much higher amount of RavZ molecules because it is driven by stoichiometric binding between RavZ and LC3-PE. The proteolytic function of RavZ drives turnover of LC3-PE molecules, so that less than stoichiometric amounts of RavZ molecules are needed. It is possible that the extraction model may be present in other host-pathogen interactions, such as *Shigella* effector IpaJ with N-myristoylated Arf GTPases and *Yersinia* effector YopT with prenylated RhoA GTPases, since these effectors also recognize the lipid group (*Burnaevskiy et al., 2013*, *2015*; *Shao et al., 2002*, *2003*).

We have demonstrated that the LIR2 motif (residue 27–32) plays an essential role in RavZ activity and RavZ:LC3 interaction. Interestingly, the LIR motif at the N-terminal tail of Atg4B is involved in regulation of Atg4B activity. Binding of the LIR motif with the second non-substrate LC3 molecule leads to an open conformation, which is required for deconjugation of LC3. Atg4B lacking the N-terminal tail shows higher processing activity (*Satoo et al., 2009*). It is not clear whether such interaction occurs in vivo. However, the LIR2 motif of RavZ is required for initial recognition of LC3. The LIR-LC3 binding constitutes the major interaction between RavZ and LC3 before extraction of the PE moiety from the membrane. Therefore, LIR2 is involved in recognizing and orienting the LC3 molecule to facilitate subsequent extraction and cleavage of LC3-PE. According to this model, it is not surprising that RavZ does not process PE-peptides containing C-terminal amino acid residues of LC3 and the Rab7-CQETFG-PE chimeric protein, due to the lack of LIR binding. In contrast, *Shigella* effector IpaJ and *Yersinia* effector YopT can process the lipidated peptides containing an N-myristoylated glycine and a C-terminally prenylated polybasic sequence, respectively, suggesting a unique manner of recognizing substrate for RavZ (*Burnaevskiy et al., 2013*, *2015*; *Shao et al., 2002*, *2003*). Based on this finding, we are able to inhibit RavZ activity by the LIR2 peptide (IC$_{50}$ = 43 µM) (*Figure 5—figure supplement 1B*), suggesting that suppression of LIR-LC3 binding could be beneficial for attenuating inhibition of host autophagy by *L. pneumophila*.

Through structural alignment with yeast lipid transfer proteins and mutagenesis studies, we have identified the LBS of RavZ, involving α2, α3, α4, the α4–β9 loop and β sheets (β6–β9). The LBS is required for both extraction and cleavage of LC3-PE. Therefore, LIR2:LC3 and LBS:PE interactions may constitute two major binding interfaces for RavZ:LC3-PE complex, which place the C-terminal tail of LC3 at the correct position of the active site of RavZ for cleavage. However, such a LBS is not observed in Atg4B (*Figure 5—figure supplement 2C*), consistent with the observation that Atg4B has no specificity toward structures C-terminal to the scissile bond. This could be one of the explanations for the distinct modes of action of Atg4 and RavZ. The fold of the N-terminal catalytic domain of RavZ is closely related to cysteine proteases in the ubiquitin-like (Ubl)-specific protease (Ulp) family that is specific for de-conjugating Ubl proteins (*Mossessova and Lima, 2000*; *Shen et al., 2005*). A similar LBS fold is also found in Ulp proteins (NEDP1, PBD: 2BKR), but this fold is lack of the α3 loop (*Figure 6—figure supplement 1A*). It should be noted that the LBS of RavZ is closed without showing a binding pocket for lipid, whereas a clear hydrophobic binding cavity that can accommodate fatty acid chains is observed in free Sec14 structures (*Figure 6—figure supplement 2B*). Because the PE moiety involves interaction with residues located deep in the LBS fold, there must be a conformational change to open the lipid-binding site, for example, by outward movement of α4. Because α4 is connected to β9 via the highly dynamic α3 loop, such a movement could be made possible. These findings are in keeping with the important role of the α3 loop in RavZ activity (*Figure 6C–6F*; *Figure 3—figure supplement 1F and G*). The hydrophobic and dynamic features of α3 are in agreement with this proposed function. The association of α3 with the membrane, as shown previously (*Horenkamp et al., 2015*), may facilitate initial interaction of α3 with the conjugated PE in the membrane. Subsequently, the lipid-binding site of RavZ is open and able to accommodate the PE moiety that is dug out of the membrane by α3, which moves toward the LBS. The α3 may serve as a lid for the lipid-binding pocket to confer the binding of fatty acid chains.

Intracellular molecules with analogous feature, that is extraction of lipidated proteins, are GDP-dissociation inhibitors (GDIs) and the GDI-like molecule PDEδ, which serve as recycling factors for prenylated Ras GTPase family proteins (Ras, Rho and Rab) between membranes and the cytosol (*Ismail et al., 2011*; *Rak et al., 2003*; *Tnimov et al., 2012*). Interestingly, the lipid-binding site of isolated RabGDI is also closed and undergoes a conformational change involving an outward movement of an α helix to open the LBS for accommodating the prenyl moiety (*Rak et al., 2003*). The opening of the LBS is induced by binding to a Rab molecule rather than the prenyl group per se (*Ignatev et al., 2008*; *Zhao et al., 2016*). Therefore, RavZ has evolved a GDI-like mechanism to extract LC3-PE from membranes. Taken together, we propose a working model of RavZ function as shown in *Figure 7*.

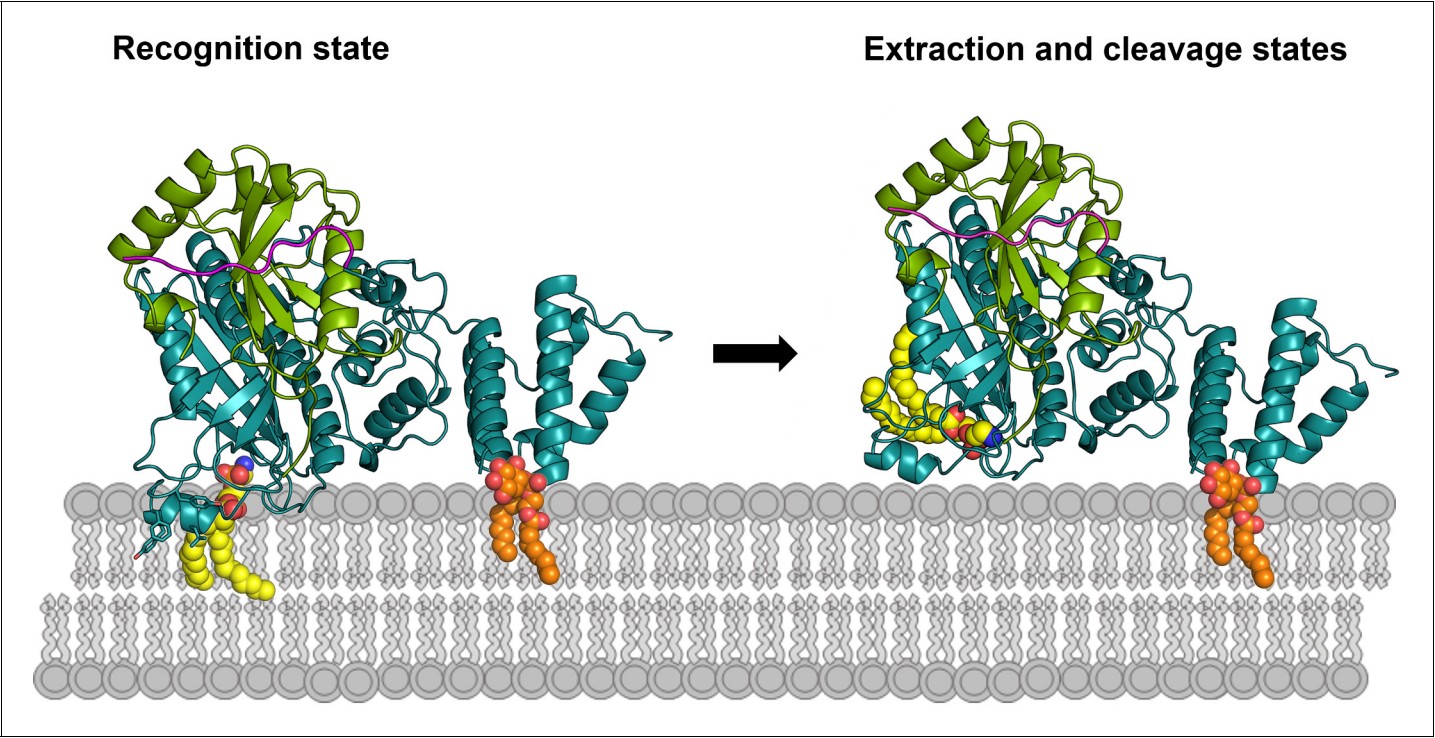

**Figure 7.** Working model of RavZ-mediated LC3-PE deconjugation on the membrane. RavZ (deep teal) recognizes LC3 molecule (green) on the membrane *via* its LIR2 motif (magenta). RavZ targets to autophagosome membrane by interaction of its C-terminal domain with PI3P (orange) and association of the α3 helix (the helix inserted into the membrane) with membranes. α3 facilitates extraction of the PE moiety (yellow) from the membrane and docking of the fatty acid chains into the lipid-binding site of RavZ. The interaction of the PE moiety with LBS and the LIR2:LC3 binding orient the C-terminal tail into the active site for cleavage. RavZ:LC3 complex was generated by molecular docking using ZDOCK server: an automatic protein docking server (http://zdock.umassmed.edu). The LC3$^{1-120}$ structure from Atg4B:LC3 complex (PDB: 2Z0D) was docked onto the structure of RavZ$^{20-502}$ (residues 48–432). The C-terminal residues (115-120) of LC3$^{1-120}$ and catalytic residues (C258, H176 and D197) located in the active site of RavZ were selected as the binding residues. The model was selected from top 10 scoring models. Binding of N-terminal LIR2 loop of RavZ with LC3 was generated by superimposition of the LIR2:LC3 structure onto LC3 in the docking structure.

## Materials and methods

### Cell lines, chemicals and antibodies

MCF7 cells (HTB-22) and HeLa (CCL-2) cells were obtained from ATCC. Human microtubule-associated protein light chain 3B (LC3) were cloned into pEGFP-C1 vector (Clontech) and pEGFP-C1-LC3 plasmid was used to generate human MCF-7 stable cell lines by single-cell Fluorescence-Activated Cell Sorting (FACS). They were tested negative for mycoplasma contamination. The cells were grown in minimum essential medium (MEM) (Sigma-Aldrich Cat# M4655) supplemented with 10% fetal bovine serum (FBS), 1% sodium pyruvate and 1% non-essential amino acids (NEAA). G418 (200 μg/mL) for the stable cell line. Cell lines were cultured at 37°C in 5% $CO_2$. For starvation, cells were first washed with PBS three times and incubated in EBSS (Sigma Cat# E3024) for 2 hr.

Chemicals for peptide and compound synthesis were obtained from Acros, Aldrich, Avanti, Fluka, Santa Cruz or Novabiochem and used without further purification.

Antibodies used in the study were rabbit anti-LC3B (Cell Signaling Technology Cat# 2775, RRID: AB_915950), anti-Cherry (Abcam Cat# ab125096, RRID:AB_11133266) and mouse anti-*β*-actin (Millipore Cat# MAB1501, RRID:AB_2223041). HRP-conjugated secondary antibodies used for WB were anti-mouse (Dako Cat# P022602, RRID:AB_579516) and anti-rabbit (Millipore Cat# AQ132P, RRID: AB_92785).

## Chemical synthesis

Peptides were synthesized employing an Fmoc-based solid phase peptide synthesis strategy using 2-chlorotrityl chloride resin. Protected peptide (0.016 mmol) was lipidated by 1-hexadecanol (16C), 1,2-dipalmitoyl-sn-glycero-3-phosphoethanolamine (DPPE) or 1,2-dihexanoyl-sn-glycero-3-phosphoethanolamine (DHPE). All peptides were characterized by high-resolution mass spectra (HR-MS) (*Supplementary file 1*). HR-MS were measured on a Thermo Orbitrap coupled to a Thermo Accela HPLC system using electrospray ionization (ESI).

The small nucleophiles, ethanolamine **1** and phosphoethanolamine **2** were purchased from Sigma. Other small nucleophiles, glycerophosphoethanolamine **8** and diacetyl glycerophosphoethanolamine **10** were synthesized as shown in *Figure 1—figure supplement 1B*. Subsequently, these compounds were characterized by NMR and HR-MS (*Supplementary file 2*). $^1$H NMR spectra were recorded on a Varian Mercury Plus 300 MHz spectrometer (300 MHz), a Bruker DRX 400 (400.13 MHz) spectrometer. NMR spectra were calibrated to the solvent signals of DMSO-$d_6$ ($\delta$ = 2.50 and 40.45 ppm). All reactions were carried out under an inert atmosphere in dry solvents unless otherwise noted.

## Synthesis of LC3 proteins containing different C-terminus modification

The lipidated proteins were achieved by performing express protein ligation (EPL) of recombinant LC3-thioester proteins with synthetic lipidated peptides according to our previous method (*Yang et al., 2013*). LC3 proteins containing soluble fragments of PE lipid were produced using direct aminolysis strategy of protein thioester by a small-molecule nucleophile (*Payne et al., 2008*; *Yi et al., 2010*).

Ligation of the DPPE (16:0) or 16-carbon hexanoyl chain modified peptide with MBP-LC3$^{1-114}$ thioester was performed as previously reported (*Yang et al., 2013*). The ligation of DHPE (6:0) modified peptide with MBP-LC3$^{1-114}$ thioester was performed in absence of the detergent. After incubation overnight at room temperature, the ligated product was purified by size exclusion chromatography (Superdex 200 10/300 GL).

Aminolysis of LC3$^{1-120}$-thioester by small-molecule nucleophile was carried out by the following procedure. 50 µM LC3$^{1-120}$-MESNA thioester was incubated with 0.5 M small molecule nucleophiles in the buffer (100 mM NaH$_2$PO$_4$, pH 7.2, 50 mM NaCl, 100 mM MPAA) for 48 hr at 4°C. The reaction was subjected to LC-MS analysis. Finally, the solution was dialyzed against dialysis buffer to remove unreacted compound.

## Expression and purification

Plasmids containing RavZ fragments or LC3 proteins were transformed into *E.coli* BL21 (DE3)-LIR cells. Protein expression was induced with 0.2–0.4 mM IPTG and carried out at 20°C overnight. Purification was performed with Äkta prime plus chromatography purification system (GE Healthcare Life Sciences). The cells were harvested and resuspended in breaking buffer containing 1X protease cocktail (Roche Life Science Cat# 05056489001) or 1 mM PMSF. Cells were then lysed by a Microfluidizer (Microfluidics). All proteins used in this study contain affinity tags (MBP, GST or chitin-binding domain (CBD)) with an extra 6xHis tag fused to its N-terminus. The proteins were initially purified by Ni-NTA-affinity purification using HisTrap HP column (GE Healthcare Life Sciences) eluted with gradient of 0–100% of 500 mM imidazole. To release the protein from the affinity tags, the fusion proteins carrying precision protease or TEV protease cleavage site were cleaved by the corresponding proteases overnight at 4°C. For the chitin fusion proteins with intein tag were cleaved by adding powdered MESNA to a final concentration of 0.5 M and incubation overnight at room temperature to produce the protein thioesters. The cleaved tags were removed with HisTrap HP column, proteins were further purified with size exclusion chromatography using HiLoad 16/60 Superdex 200.

For analytical gel filtration analysis of the complex formation. The experiments were performed on Superdex 200 10/300 GL equipped with ÄKTAFPLC system. Protein samples were previously purified with size exclusion chromatography as a last purification step. Complex formation was done by mixing each of RavZ fragment and LC3$^{1-119}$ with molar ratio of 1:1.5 and incubated at 4°C overnight. 500 µL of proteins or protein complexes were injected to pre-equilibrated column with buffer containing 50 mM HEPES pH 7.5, 50 mM NaCl and 2 mM DTE. The size exclusion chromatography was performed with a flow rate of 0.5 ml/min for 35 min. The standard protein markers containing

(thyroglobulin (670 kDa), globulin (158 kDa), ovalbumin (44 kDa), myoglobin (17 kDa) and vitamin B12 (1.3 kDa)) were run on the same column under identical condition.

## Crystallization and structural determination

For crystallization screening, proteins were screened for crystallization conditions with various crystallization screening kits from QIAGEN. Protein and screening buffer were automatically mixed in ratio 1:1 with 0.1 µL of each by mosquito in crystallization screening plate. The crystallization plates were incubated in the Rock Imager (Formulatrix), an automated imaging system at 277.15 K and 293.15 K. RavZ$^{1–430}$/LC3B$^{1–119}$ complex was crystallized in 0.17 M ammonium acetate, 0.085 M tri sodium citrate pH 5.6, 25.5% PEG4000 and 15% glycerol at 277.15 K. RavZ$^{1–487}$ and RavZ$^{20-502}$ were crystallized in 16% PEG3350, 0.2 M BaCl$_2$ and 0.1 M MES pH 5.6 at 277.15 K. LIR2-LC3$^{1–119}$ fusion protein was crystallized in two conditions, as indicated with low-salt and high-salt. The low-salt condition contains 0.1 M citric acid anhydrous pH 4.0 and 1.6 M ammonium sulfate at 293.15 K and high-salt condition contains 0.1 M sodium acetate pH 4.5 and 3 M NaCl at 277.15 K. The crystallization conditions were further optimized by refining concentration of precipitant, salt and pH.

The data were collected at 100 K at PXII-XS10SA beamline in the Swiss Light Source (SLS) Villingen. Data was indexed, processed and scaled with XDS (*Kabsch, 1993*). RavZ (PDB: 5CQC) and LC3B (PDB: 2ZOD) were used as molecular replacement search models. Molecular replacement and structural refinement were done with PHENIX (*Adams et al., 2010*) and rebuilt in COOT (*Emsley and Cowtan, 2004*). Validation was done with Molprobity (*Chen et al., 2010*). X-ray data collection and refinement statistics are listed in *Supplementary file 4*.

## Enzyme kinetics

### In vitro LC3-PE cleavage assay

7 µM LC3-DPPE (16:0) or LC3-DHPE (6:0) was incubated with 0.7 µM RavZ at 37°C. At indicated time points, 10 µL sample was taken and subsequently mixed with 4x SDS sample buffer to quench the reaction. Samples were analyzed by SDS-PAGE.

### Kinetic analysis of LC3-PE cleavage by RavZ

Different concentrations of LC3-DPPE (16:0) or LC3-DHPE (6:0) (0.7, 1.1, 2.1, 4.2, 7.0, 10.5, 14.0, 21.0 and 28.0 µM) were mixed with 0.7 µM RavZ concentration. LC3-DPPE (16:0) was incubated with RavZ for 8 min and LC3-DHPE (6:0) for 140 min. The reaction was quenched by SDS loading buffer and samples were analyzed by SDS-PAGE.

### Inhibition assay of LIR2 peptide

LC3-PE (7 µM) was incubated with different concentration of the LIR2 peptide for 1 hr on ice. The mixture was subjected to RavZ (0.7 µM) treatment for 10 min at 37°C and then analyzed by SDS-PAGE. IC$_{50}$ value was calculated using four-parameter logistic function by Origin.

### Quantitative analysis

To quantitatively analyze the SDS-PAGE results, the intensity of the bands on the polyacrylamide gel was measured by determination of the optical density (OD). A density profile was created and the integral was calculated by ImageJ. The percentage of substrate (LC3-PE) was defined for each time point as (*Equation 1*):

$$Substrate\ (\%) = \frac{OD_{Substrate}}{OD_{Substrate} + OD_{Product}} \times 100\% \tag{1}$$

The percentage of substrate was plotted against the RavZ incubation time and the half-time (t$_{1/2}$) of cleaved PE from LC3 was calculated by exponential fitting using *Origin*.

The initial velocity $V$ (µM/s) defined as the change in the concentration of cleaved product (LC3$^{1–119}$) was plotted against the concentration of the substrate LC3-PE, $S$ (µM). The maximal velocity $V_{max}$ and the Michaelis constant $K_m$ were obtained by fitting data to Michaelis-Menten equation. The catalytic constant $k_{cat}$ was defined as ratio between $V_{max}$ and enzyme concentration and the catalytic efficiency as ratio between $k_{cat}$ and $K_m$.

## Mass analysis of modified proteins and cleaved product

LC-MS analysis was performed on an Agilent 1100 series chromatography system equipped with an LCQ electrospray mass spectrometer (Finnigan, San Jose) using Jupiter C4 columns (5 µm, 15 × 0.46 cm, 300 Å pore-size) from Phenomenex (Aschaffenburg, Germany). For LC-separations a gradient of buffer B (0.1% formic acid in acetonitrile) in buffer A (0.1% formic acid in water) with a constant flow-rate of 1 mL/min was employed. Upon sample injection, a ratio of 20% buffer B was kept constant for 4 min. Elution was achieved using a linear gradient of 30–80% buffer B in buffer A for 5–15 min followed by a steep gradient (70–90% buffer B) for 15–17 min. The column was extensively flushed for 17–19 min with 90–10% buffer B. Data evaluation was carried out using the Xcalibur software package and MagTran software programs was used for deconvolution of ESI mass spectra (*Supplementary file 3*).

## Binding affinity measurements

### Binding measurements using fluorescence polarization

Fluorescence polarization was performed on Tecan safire II. The LC3 was introduced a single cysteine at the N-terminus for labeling with Tide Fluor 3 (TF3) maleimide dye. 6.25 µM labeled LC3 was titrated with RavZ in the buffer containing 25 mM HEPES, pH 7.2, 25 mM $NaCl_2$ and 2 mM DTE. The measurement was carried out in the 384-well plate. Measuring was set in a polarization mode with excitation and emission wavelength at 530 nm and 584 nm, respectively. For binding measurement of FITC-LIR2 peptide with LC3 and its variants as well as GABARAPL1 and GATE16, 2.5 µM FITC-LIR2 was used with the excitation and emission at 470 nm and 525 nm, respectively.

Non-linear binding curves were obtained from plotting the polarization signal against log concentration of the titrants. $K_d$ values were obtained by fitting binding curves with a quadratic equation (*Equation 2*) using GraFit5.

$$F = F_{min} + \left( K_d + A_0 + E_0 - \sqrt{(K_d + A_0 + E_0)^2 - 4A_0E_0} \right) \frac{F_{max} - F_{min}}{2A_0} \qquad (2)$$

where $A_0$ and $E_0$ are the total concentrations (free and bound) of labeled ligand A (LC3) and titrant E (RavZ), respectively. $F$ is the experimentally observed fluorescence polarization signal, $F_{min}$ is the initial signal at $E_0 = 0$, $F_{max}$ is the final signal at $E_0 = \infty$.

### Binding measurement using isothermal titration calorimetry (ITC)

The titrant LIR2 peptide with a concentration of 700 µM was injected into 70 µM LC3 with 20 injections at 33°C with the duration and spacing times of 4 s and 60 s, respectively. The experiments were done in the buffer containing 50 mM HEPES pH 7.2, 50 mM NaCl and 2 mM 2-mercaptoethanol. Titration of the buffer into LC3 was used as the blank for background subtraction. The heat release was plotted against the concentration of titrant and subtracted with background, yielding the binding curve. The $K_d$ values of the binding were obtained by fitting the final binding curves with a single binding mode using Microcal software.

### Binding measurement using MST

The inactive RavZ (C258A) was labeled with TF3 dye and 2.5 µM of the labeled RavZ (C258A) was added into titrated concentration of MBP-LC3-PE (16:0) or MBP-LC3$^{1–119}$ (41.5 nM to 5.07 nM) in a buffer containing 20 mM HEPES pH 7.2, 25 mM $NaCl_2$ and 2 mM DTE. The experiments were done at 28°C. The samples were filled into standard capillaries and measured with a 34% LED and 60% IR-Laser with laser-on time of 30 s and laser-off time of 5 s. The movement of the molecules along the temperature gradient results in a reduction of fluorescence. Plotting the reduction of fluorescence vs. the concentration of the titrants yields a non-linear binding curve. The binding curves were fitted with the nonlinear solution of the law of mass action.

## Protein secondary structure estimation using circular dichroism

CD spectroscopy was acquired using a Jasco J-815 CD spectrometer equipped with a JASCO PTC-423S temperature controller. Proteins were diluted to 0.125 mg/ml in 50 mM phosphate buffer pH 7.5 containing 50 mM NaCl and 2 mM DTE. Protein samples were filled in 0.1 cm quartz cuvette and

scan from 195 nm to 260 nm at 25°C, scanning speed was 20 nm/min and CD spectra were accumulated three times. Firstly, CD spectra of protein samples were subtracted by spectra of buffer and smoothed using a FFT filter. The unit of millidegree ($\theta$) was conversed to mean residue molar ellipticity $[\theta]_{MRW,\lambda}$. The ratio of secondary structure elements was calculated with the software CDNN. The database consisting of 33 reference proteins was used in the deconvolution analysis.

## Imaging and immunoblotting

For imaging, MCF7 cells stably expressing GFP-LC3 or HeLa cells were plated at a density of $5.0 \times 10^4$ cells per well on μ-Slide 4 Well (Ibidi). GFP-LC3 stable cells were transfected with 250 ng of mCherry-RavZ constructs using X-treme GENE HP DNA transfection reagent. For co-transfection, HeLa cells were co-transfected with each 125 ng of GFP-LC3 and mCherry-RavZ constructs. After 20 hr, live cell imaging was performed in MEM without phenol red (Thermo Fisher Cat# 51200046) or EBSS (Sigma Cat# E3024) by using an inverted confocal microscope Leica TCS SP2 or SP5 AOBS equipped with a 63×/1.4 HCX Plan Apo oil immersion lens and a temperature-controlled hood at 37°C and 5% $CO_2$. Quantification of the area of GFP-LC3 puncta was performed through Analyze Particles function of ImageJ.

For immunoblotting, GFP-LC3 stable cells were plated at a density of $5.0 \times 10^5$ cells per well in a 6-well plate. Cells were transfected with 1 μg of each plasmid using X-treme GENE HP DNA transfection reagent (Roche Life Science Cat# 06365244001). For immunoblotting, total cell lysates were prepared by adding cells to RIPA buffer (50 mM Tris pH 7.8, 150 mM NaCl, 1% Triton X-100, 1% sodium deoxycholate and 0.1% sodium dodecyl sulfate (SDS)). Equal amount of proteins was resolved by SDS-PAGE, transferred to PVDF membranes, and incubated with primary antibody overnight at 4°C. After washing 3 times with TBST buffer, membranes were incubated with secondary antibodies conjugated with horseradish peroxidase for 60 min. Signals were visualized with ECL Prime Western Blotting Detection Reagent(Amersham Cat# 10600001) or SuperSignal Western Blot Enhancer (Thermo Fisher Cat# 34095) using UltraCruz Autoradiography Films (Santa Cruz Cat# sc-201696).

## LC3-PE membrane extraction assay

MCF7 cell membranes were prepared as described (*Pylypenko et al., 2006*; *Rak et al., 2003*). GFP-LC3 stable cells were starved in EBSS for 2 hr to induce autophagy. The cells were collected and resuspended in the fractionation buffer (FB buffer, 250 mM Sucrose, 20 mM HEPES (7.4), 10 mM KCl, 1.5 mM $MgCl_2$, 1 mM EDTA, 1 mM DTT and protease inhibitor cocktail) on ice for 30 min. The cell solution was homogenized 80 times using Dounce homogenizer. The cell lysate was cleared by centrifugation at 540 g for 5 min. The extract was then loaded onto a 1 mL cushion of sucrose (60% in RB), and centrifuged at 100,000 g for 1 hr in a Optima MAX-XP ultracentrifuge equipped with rotor TLA_100.4 (Beckman Coulter). The buffer–sucrose interface was collected in a minimal volume, and the protein concentration was determined by the Bradford assay.

Different concentrations of RavZ$^{C258A}$ protein were incubated with the membrane fraction (20 μg of membrane protein) in the final assay volume of 100 μL at 37°C for 2 hr. 500 μL of FB buffer was added to each reaction and mixed. Diluted assay mixtures were centrifuged for 1 hr at 100,000 g using the ultracentrifuge equipped with rotor TLA_100.1 (Beckman Coulter). Precipitation of the supernatant was done using TCA/DOC (trichloroacetic acid/sodium deoxycholate) protocol. Briefly, DOC (final concentration 125 μg/ml) and TCA (final concentration 6%) were added to the protein fraction, subsequently. The resulting solution was mixed and incubated on ice for 20 min, and then was centrifuged 15 min in max speed (14,680 rpm). The pellet was washed by cold acetone and dissolved in 1X SDS sample buffer. The samples were resolved on 13% SDS–PAGE gel. Membrane-associated and extracted endogenous LC3-II and GFP-LC3-II were visualized by immunoblotting with anti-LC3 antibody.

For detection of cytosolic LC3-PE level in the RavZ$^{C258A}$ transfected cells, GFP-LC3 stable cells were plated to 6 cm dish at a density of $1.6 \times 10^6$ cells per dish. The cells were transfected with 3 μg of plasmid pmCherry-RavZ$^{C258A}$ using X-treme GENE HP DNA transfection reagent. The pmCherry vector also was used as a control. After 24 hr, cells were starved and collected, and resuspended in the FB buffer on ice for 30 min. The cell solution was homogenized 80 times using Dounce homogenizer. The cell lysate was cleared by centrifugation at 540 g for 5 min. The extract

(total protein fraction) was then centrifuged at 100,000 g for 1 hr in an Optima MAX-XP ultracentrifuge equipped with rotor TLA_100.1 (Beckman Coulter). The supernatant (cytosolic fraction) was collected and the pellet (membrane fraction) was resuspended in the RIPA buffer. The protein concentration was determined by the Bradford assay. LC3 proteins were visualised by immunoblotting with anti-LC3 antibody.

## Quantification and statistical analysis

Results were expressed as mean ± standard deviation of means (SD). The data were assessed by one-way analysis of variance (ANOVA) followed by the Fisher LSD's post hoc comparisons. In all statistical comparisons, differences with $p < 0.05$ were considered significance. Statistical analysis was performed in Origin 9.

## Accession codes

Atomic coordinates and structure factors have been deposited in the Protein Data Bank under the following accession codes 5MS2 (RavZ$^{1-431}$/LC3B), 5MS5 (low-salt RavZ LIR2-LC3B), 5MS6 (high-salt RavZ LIR2-LC3B), 5MS7 (RavZ$^{20-502}$) and 5MS8 (RavZ$^{1-487}$).

## Acknowledgements

We thank Ingrid Vetter and Emerich Mihai Gazdag for the assistance in X-ray structure determination and fruitful discussion, and Inken Hacheney for performing experiments shown in *Figure 2—figure supplement 2B and C* (right panel). This work was supported by Deutsche Forschungsgemeinschaft, DFG (grant No: SPP 1623), Behrens Weise Stiftung and European Research Council, ERC (ChemBioAP) to Y.W.W. and by the Introduction of Innovative R and D Team Program of Guangdong Province (2009010058) to A.Y. We thank Roger Goody for proof reading of the manuscript. We acknowledge the staff of Beamline X10SA at the Paul Scherrer Institute, and the X-ray communities at the Max-Planck-Institute (MPI) Dortmund for technical support.

## Additional information

### Funding

| Funder | Grant reference number | Author |
|---|---|---|
| The Introduction of Innovative R&D Team Program of Guangdong Province | 2009010058 | Aimin Yang |
| Deutsche Forschungsgemeinschaft | SPP 1623 | Yao-Wen Wu |
| European Research Council | ChemBioAP | Yao-Wen Wu |
| Behren-Weise Stiftung | | Yao-Wen Wu |

The funders had no role in study design, data collection and interpretation, or the decision to submit the work for publication.

### Author contributions

AY, SP, Data curation, Formal analysis, Investigation, Methodology, Writing—original draft, Writing—review and editing; Y-WW, Conceptualization, Formal analysis, Supervision, Funding acquisition, Investigation, Methodology, Writing—original draft, Project administration, Writing—review and editing

### Author ORCIDs

Aimin Yang, http://orcid.org/0000-0002-2240-5549
Supansa Pantoom, http://orcid.org/0000-0001-9630-1999
Yao-Wen Wu, http://orcid.org/0000-0002-2573-8736

# Additional files

## Supplementary files

• Supplementary file 1. Mass characterization of peptides by HR-MS.

• Supplementary file 2. Chemical synthesis of compound 8 and 10.

• Supplementary file 3. Mass characterization of modified LC3 proteins by ESI-MS.

• Supplementary file 4. Data collection and refinement statistics.

## Major datasets

The following datasets were generated:

| Author(s) | Year | Dataset title | Dataset URL | Database, license, and accessibility information |
|---|---|---|---|---|
| Supansa Pantoom, Ingrid Vetter, Yao-Wen Wu | 2017 | Crystal structure of the Legionella pneumophila effector protein RavZ in complex with human LC3B | http://www.rcsb.org/pdb/explore/explore.do?structureId=5MS2 | Publicly available at the RCSB Protein Data Bank (accession no: 5MS2) |
| Supansa Pantoom, Ingrid Vetter, Yao-Wen Wu | 2017 | Crystal structure of the legionella pneumophila effector protein RavZ_20-502 | http://www.rcsb.org/pdb/explore/explore.do?structureId=5MS7 | Publicly available at the RCSB Protein Data Bank (accession no: 5MS7) |
| Supansa Pantoom, Ingrid Vetter, Yao-Wen Wu | 2017 | Low-salt structure of RavZ LIR2-fused human LC3B | http://www.rcsb.org/pdb/explore/explore.do?structureId=5MS5 | Publicly available at the RCSB Protein Data Bank (accession no: 5MS5) |
| Supansa Pantoom, Ingrid Vetter, Yao-Wen Wu | 2017 | High-salt structure of RavZ LIR2-fused human LC3B | http://www.rcsb.org/pdb/explore/explore.do?structureId=5MS6 | Publicly available at the RCSB Protein Data Bank (accession no: 5MS6) |
| Supansa Pantoom, Ingrid Vetter, Yao-Wen Wu | 2017 | Crystal structure of the legionella pneumophila effector protein RavZ_1-487 | http://www.rcsb.org/pdb/explore/explore.do?structureId=5MS8 | Publicly available at the RCSB Protein Data Bank (accession no: 5MS8) |

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
