## [Decision Letter]

Thank you for submitting your article "Elucidation of the *Legionella*effector RavZ against host autophagy using semisynthetic LC3 proteins" for consideration by *eLife*. Your article has been favorably evaluated by Ivan Dikic (Senior Editor) and three reviewers, one of whom, Volker Dötsch (Reviewer #1), is a member of our Board of Reviewing Editors. The following individuals involved in review of your submission have agreed to reveal their identity: Kim Orth (Reviewer #2); Hubert Hilbi (Reviewer #3).

The reviewers have discussed the reviews with one another and the Reviewing Editor has drafted this decision to help you prepare a revised submission.

Summary:

Yang et al. analyze the mode of action of the *Legionella* protein RavZ. This translocated effector protein has previously been shown to irreversibly deconjugate Atg8/LC3-phosphatidylethanolamine (PE), thus inhibiting the autophagy pathway. The authors use semi-synthesis approaches to produce LC3 proteins modified with different PE derivatives. These substrates are then employed to assess binding, catalytic activity and structure-function relationships of the RavZ-LC3 complex. The study shows that RavZ (i) specifically cleaves LC3-DPPE and LC3-DHPE in vitro, (ii) extracts LC3-PE from membranes prior to cleavage, and (iii) binds to LC3 through an N-terminal LIR motif, which is composed of conserved aromatic/hydrophobic amino acids and required for substrate cleavage. Moreover, (iv) the crystal structure of a RavZ:LC3 complex indicated that RavZ forms a lipid binding site homologous to the eukaryotic lipid transfer protein Sec14, and (v) the functional relevance of this lipid binding groove was tested by mutational analysis and in vitro LC3-PE cleavage assays.

All reviewers agreed that this is a very interesting study. However, several points were raised that are listed below. In particular, certain information is missing from some of the figure legends and the model should be improved.

Essential revisions:

1) "Apo-protein/-enzyme" is canonically used to designate a protein/enzyme lacking its non-proteinaceous co-factor. Accordingly, the terms "apo-structure", "apo-RavZ structure" and "apo-Sec14 structure", which are used in the manuscript to describe proteins/structures in absence of a (lipo-)protein binding partner, seems misleading. Please correct.

2) Figure legend 2 and Figure legend 2 supplements:

Legend to Figure 2: the acronyms of the LC3-derivatives should be defined: "EA", "pEA", "GpEA", "DAGpEA", "DPPE" and "DHPE".

Figure 2—figure supplement 1: The Atg4B gel is the same one used for the Yang 2013 paper. You can see the same horizontal line on the top left portion of the gel. It would be better to either repeat the experiment or just cite the results in the text, instead of reusing the figure.

Figure 2—figure supplement 2: The cleavage product for the LC3(1-120)-PE(16:0) seems really faint in this gel. Was it possible to accurately quantitate it and was the experiment done in replicates?

3) Figure 3: The panel for extracted untagged-LC3 is missing here.

4) In Figure 4—figure supplement 1: The chromatograms of the complex seem to indicate the existence of several additional peaks (species). What are those peaks?

5) Figure 5 legend: Is the gray structure in B apo-RavZ? This is not indicated in the legend. Also, the color for the PI3P domain in A is listed as deep-pearl in the legend instead of teal.

6) Figure legend 6 and Figure legend 6 supplement:

Figure 6: Please number the secondary structure elements in the Sec14, the text is impossible to follow without these. The identity of the pink ligand is not indicated anywhere in the figure or the legend.

In addition in legend to Figure 6: define "lipid-binding site (LBS)".

Figure 6—figure supplement 1: It is not clear at all why NEDP1 is used in this alignment and it is not discussed at all in the text. Please explain why the change from Sec14 to NEDP1 and indicate the PDB ID code for NEDP1. It would also be very helpful to number secondary structure elements in A and C as they are discussed in the text.

7) Figure legend 7 and Figure legend 7 supplement:

Figure 7 legend: A citation should be provided for the ZDOCK server. Also, no details about how the docking was performed are offered here or in the Methods. It would be good to know whether full length or truncated structures were used as input for the docking program.

In addition, omit "sphere" – (orange), (yellow).

The model in Figure 7 oversimplifies the mechanism. The effector is delivered to the host and is targeted to the autophagic membrane via PI3P binding. Once bound, by lateral movement, RavZ finds an LC3-PE molecule, extracts the lipid and cleaves its substrate. LC3 is a soluble protein but what happens to the lipid? As drawn in your model the whole complex becomes soluble and there is no longer PIP3 binding? For the third figure, membrane should be underneath the complex, if it is rotated. RavZ will remain on the membrane via its binding to PI3P – in search of another substrate.

8) Methods section 8.2: Binding measurement using isothermal titration calorimetry (ITC). Why was a titration of buffer into RavZ used to measure background? Titration of buffer into LC3 or LIR2 peptide into buffer would be logical choices to control for background. Also, it would be good to show the thermograms and isotherms in Figure 4—figure supplement 1.

9) In the section "RavZ extracts LC3-PE from membranes" the reader does not know that membranes are being tested unless they read the methods. Be more explicit in the explanation of your experiments.

10) Identification of the potential binding site of the lipid is based on structural comparison and mutational analysis. This is however indirect. A direct identification would be the crystal structure of RavZ with lipidated LC3 or at least a binding study via ITC with lipidated LC3 showing that modified LC3 binds stronger to RavZ than LC3 without lipids. Such a direct interaction study would be much stronger than the current indirect one.

Title:

The title of the manuscript seems somewhat odd: "Elucidation" of what? Perhaps, the authors might want to consider "Elucidation of the anti-autophagy mechanism of the *Legionella*effector RavZ using semisynthetic LC3 proteins".

---

## [Author Response]

*Essential revisions:*

*1) "Apo-protein/-enzyme" is canonically used to designate a protein/enzyme lacking its non-proteinaceous co-factor. Accordingly, the terms "apo-structure", "apo-RavZ structure" and "apo-Sec14 structure", which are used in the manuscript to describe proteins/structures in absence of a (lipo-)protein binding partner, seems misleading. Please correct.*

We agree with the reviewers that using the term of Apo-protein might be misleading. Therefore, we now use “free” protein instead of Apo-protein throughout the manuscript.

*2) Figure legend 2 and Figure legend 2 supplements:*

*Legend to Figure 2: the acronyms of the LC3-derivatives should be defined: "EA", "pEA", "GpEA", "DAGpEA", "DPPE" and "DHPE".*

We have defined the acronyms of the LC3-derivatives in the legend of Figure 2 in the revised manuscript.

*Figure 2—figure supplement 1: The Atg4B gel is the same one used for the Yang 2013 paper. You can see the same horizontal line on the top left portion of the gel. It would be better to either repeat the experiment or just cite the results in the text, instead of reusing the figure.*

The biochemical experiments were repeated three times. The result on Atg4B-mediated cleavage of pro-LC3 and semisynthetic LC3-PE is now replaced by a new figure in the revised manuscript.

*Figure 2—figure supplement 2: The cleavage product for the LC3(1-120)-PE(16:0) seems really faint in this gel. Was it possible to accurately quantitate it and was the experiment done in replicates?*

The experiment was done in triplicates. The bands for lower concentrations are indeed faint. We have now redone the experiments with longer reaction time, in order to produce sufficient amount of product. The error bars are now included in the graph of quantification.

*3) Figure 3: The panel for extracted untagged-LC3 is missing here.*

We have now included the extraction of endogenous LC3-II.

*4) In Figure 4—figure supplement 1: The chromatograms of the complex seem to indicate the existence of several additional peaks (species). What are those peaks?*

The complexes were formed by mixing the purified proteins of each RavZ fragment with LC3^1-119^ in a molar ratio of 1:1.5 and incubated overnight on ice. The additional peak at later time represents uncomplexed excess LC3^1-119^, as shown in the SDS-PAGE analysis of elution fractions from gel filtration. We have now illustrated the additional peak in the gel filtration profiles and included the SDS-PAGE analysis.

*5) Figure 5 legend: Is the gray structure in B apo-RavZ? This is not indicated in the legend. Also, the color for the PI3P domain in A is listed as deep-pearl in the legend instead of teal.*

Yes, it is an apo-RavZ. It is now illustrated as free RavZ in the legend and PI3P domain is now indicated with teal.

*6) Figure legend 6 and Figure legend 6 supplement:*

*Figure 6: Please number the secondary structure elements in the Sec14, the text is impossible to follow without these. The identity of the pink ligand is not indicated anywhere in the figure or the legend.*

The secondary elements (α7-α10) of yeast Sec14 homolog Sfh1 are now labeled. The bound PE (pink ligand) is indicated in the legend.

*In addition in legend to Figure 6: define "lipid-binding site (LBS)".*

Done.

*Figure 6—figure supplement 1: It is not clear at all why NEDP1 is used in this alignment and it is not discussed at all in the text. Please explain why the change from Sec14 to NEDP1 and indicate the PDB ID code for NEDP1. It would also be very helpful to number secondary structure elements in A and C as they are discussed in the text.*

We have pointed out in the text: “The fold of the N-terminal catalytic domain of RavZ is closely related to cysteine proteases in the ubiquitin-like(Ubl)-specific protease (Ulp) family that is specific for de-conjugating Ubl proteins (Mossessova and Lima, 2000; Shen et al., 2005). A similar LBS fold is also found in Ulp proteins (NEDP1, PBD: 2BKR), but this fold is lack of the α3 loop (Figure 6—figure supplement 1).”

The secondary elements are now indicated in A and C.

*7) Figure legend 7 and Figure legend 7 supplement:*

*Figure 7 legend: A citation should be provided for the ZDOCK server. Also, no details about how the docking was performed are offered here or in the Methods. It would be good to know whether full length or truncated structures were used as input for the docking program.*

We have now added more information in the legend of Figure 7: “ZDOCK server: an automatic protein docking server (http://zdock.umassmed.edu). The LC3^1-120^ structure from Atg4B:LC3 complex (PDB: 2Z0D) was docked onto the structure of RavZ^20-502^ (residues 47-432). The C-terminal residues (115-120) of LC3^1-120^ and catalytic residues (C258, H176 and D197) located in the active site of RavZ were selected as the binding residues. The model was selected from top ten scoring models.”

*In addition, omit "sphere" – (orange), (yellow).*

Done.

*The model in Figure 7 oversimplifies the mechanism. The effector is delivered to the host and is targeted to the autophagic membrane via PI3P binding. Once bound, by lateral movement, RavZ finds an LC3-PE molecule, extracts the lipid and cleaves its substrate. LC3 is a soluble protein but what happens to the lipid? As drawn in your model the whole complex becomes soluble and there is no longer PIP3 binding? For the third figure, membrane should be underneath the complex, if it is rotated. RavZ will remain on the membrane via its binding to PI3P – in search of another substrate.*

We agree with reviewer that the third figure might be misleading. We now exclude it from the model in Figure 7.

*8) Methods section 8.2: Binding measurement using isothermal titration calorimetry (ITC). Why was a titration of buffer into RavZ used to measure background? Titration of buffer into LC3 or LIR2 peptide into buffer would be logical choices to control for background. Also, it would be good to show the thermograms and isotherms in Figure 4—figure supplement 1.*

It was a typo. It should be “LC3”. We have now corrected it. The experiments were done in triplicates (Figure 8). We include one of the ITC data in Figure 4—figure supplement 1.

Author response image 1.Interaction analysis of LIR II with LC3 using ITC technique.**DOI:**
http://dx.doi.org/10.7554/eLife.23905.023

*9) In the section "RavZ extracts LC3-PE from membranes" the reader does not know that membranes are being tested unless they read the methods. Be more explicit in the explanation of your experiments.*

We have now added some experimental details: “The membrane fraction of cells was incubated with different concentrations of RavZC258A protein. The supernatant was precipitated using TCA/DOC (trichoroacetic acid/sodium deoxycholate). The soluble proteins in supernatant and the membrane-associated proteins were visualized by immunoblotting with anti-LC3 antibody.”

*10) Identification of the potential binding site of the lipid is based on structural comparison and mutational analysis. This is however indirect. A direct identification would be the crystal structure of RavZ with lipidated LC3 or at least a binding study via ITC with lipidated LC3 showing that modified LC3 binds stronger to RavZ than LC3 without lipids. Such a direct interaction study would be much stronger than the current indirect one.*

We agree with the reviewers’ comment. We have now performed microscale thermophoresis (MST) measurements to compare the direct interaction of RavZ with LC3-PE and LC3. To make LC3-PE soluble in solution without detergent, MBP tag was left intact. The measurements showed that RavZ^C258A^ binds to MBP-LC3-PE and MBP-LC3^1-119^ with dissociation constants (K_d_) of 23 ± 4 nM and 69 ± 5 nM, respectively, which suggests that RavZ binds to lipidated LC3 in 3 times higher affinity than unlipidated LC3. Therefore, the thermodynamic driving force for RavZ extraction is modest but still favorable. We have now added the discussion in the text.

It should be noted that the function of the LBS has been demonstrated by in vitro and in vivo cleavage and extraction assay (Figure 3—figure supplement 1, Figure 6).

Author response image 2.MST measurements of binding of RavZ(C258A) with MBP-LC3^1-119^ (4 replicated experiments).**DOI:**
http://dx.doi.org/10.7554/eLife.23905.024

Author response image 3.MST measurments of binding of RavZ(C258A) with MBP-LC3-PE (4 replicated experiments).**DOI:**
http://dx.doi.org/10.7554/eLife.23905.025

*Title:*

*The title of the manuscript seems somewhat odd: "Elucidation" of what? Perhaps, the authors might want to consider "Elucidation of the anti-autophagy mechanism of the Legionella effector RavZ using semisynthetic LC3 proteins".*

We have now changed the title.